# ScaleCap: Scalable Image Captioning via Dual-Modality Debiasing

**Long Xing**[1,2]\*, **Qidong Huang**[1]\*, **Xiaoyi Dong**[2,3,†], **Pan Zhang**[2], **Yuhang Zang**[2], **Yuhang Cao**[2],
**Jinsong Li**[2,3], **Shuangrui Ding**[2,3], **Weiming Zhang**[1], **Nenghai Yu**[1],
**Jiaqi Wang**[2,†], **Feng Wu**[1], **Dahua Lin**[2,3,4]
[1]University of Science and Technology of China, [2]Shanghai Artificial Intelligence Laboratory,
[3]The Chinese University of Hong Kong, [4]CPIl under InnoHK
`{xing_long@, hqd0037@}mail.ustc.edu.cn`

## Abstract

This paper presents ScaleCap, a scalable image captioning strategy that generates comprehensive and detailed image captions. The key challenges of high-quality image captioning lie in the inherent biases of LVLMs: multimodal bias resulting in imbalanced descriptive granularity, offering detailed accounts of some elements while merely skimming over others; linguistic bias leading to hallucinated descriptions of non-existent objects. To address these issues, we propose a scalable debiased captioning strategy, which continuously enriches and calibrates the caption with increased inference budget. Specifically, we propose two novel components: heuristic question answering and contrastive sentence rating. The former generates content-specific questions based on the image and answers them to progressively inject relevant information into the caption. The latter employs sentence-level offline contrastive decoding to effectively identify and eliminate hallucinations caused by linguistic biases. With increased inference cost, more heuristic questions are raised by ScaleCap to progressively capture additional visual details, generating captions that are more accurate, balanced, and informative. Extensive modality alignment experiments demonstrate the effectiveness of ScaleCap. Annotating 450K images with ScaleCap and using them for LVLM pretraining leads to consistent performance gains across 11 widely used benchmarks. Furthermore, ScaleCap showcases superb richness and fidelity of generated captions with two additional tasks: replacing images with captions in VQA task, and reconstructing images from captions to assess semantic coverage. Code is available at `https://github.com/Cooperx521/ScaleCap`.

## 1 Introduction

In the realm of large vision language models (LVLMs) (Bai et al., 2025; 2023; OpenAI, 2023a;b; Liu et al., 2023; Dai et al., 2023), the quality and richness of image-text pairs play a pivotal role in determining the effectiveness of pre-training (Chen et al., 2023; Liu et al., 2023; Li et al., 2022). Particularly, longer and more descriptive captions have shown increasing importance in supporting fine-grained vision-language alignment, moving from early-stage captions with only a few generic words (Chen et al., 2015; Li et al., 2022) to recent efforts that generate paragraph-level, context-rich descriptions (Chen et al., 2023; Sun et al., 2024). As the field pushes toward building ever more capable foundation models (Meta AI, 2025; Grattafiori et al., 2024; Bai et al., 2023; Touvron et al., 2023), the need for vast quantities of high-quality multimodal data becomes increasingly urgent. However, relying on human annotation (Xu et al., 2024; Urbanek et al., 2024) or proprietary APIs (Dong et al., 2024; Chen et al., 2023) to produce such detailed captions proves prohibitively expensive and fundamentally non-scalable. This challenge has spurred growing interest in developing scalable captioning strategies based on open-source LVLMs, offering a more cost-effective and flexible path for constructing large-scale, high-quality captions.

---

\*Equal contribution.
†Corresponding author.

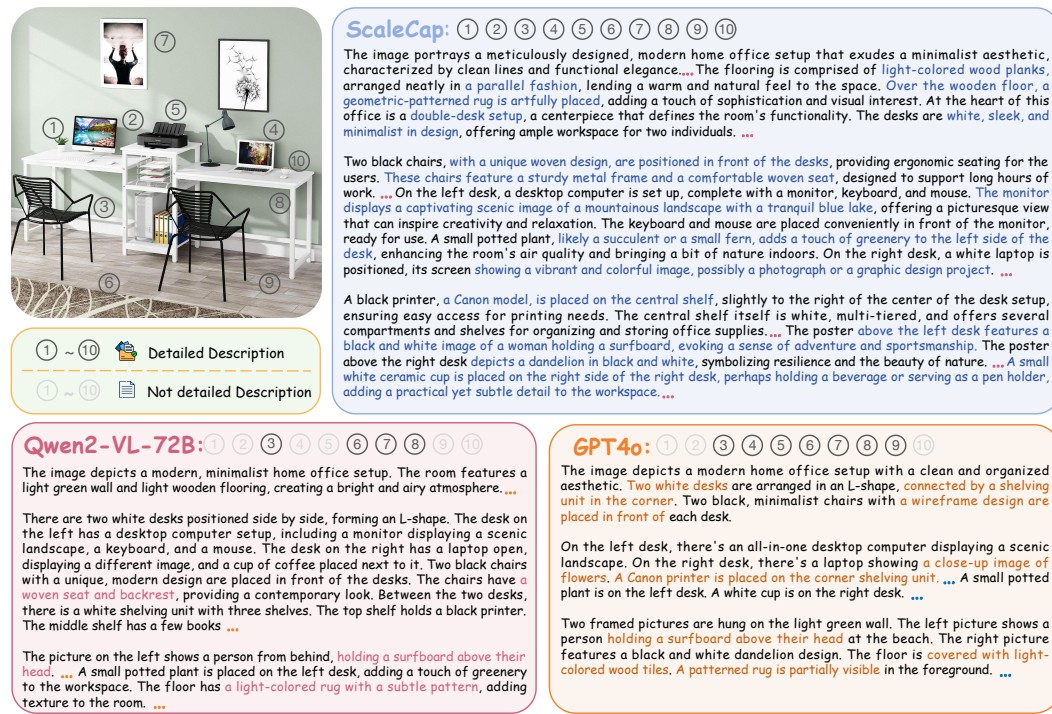

Figure 1: **Comparison between the captions** generated by our ScaleCap and those produced by other advanced VLMs. The parts of the caption that are bolded refer to the detailed descriptions of the object, while the parts that do not mention the target object are included in the ellipsis.

Despite growing interest, open-source LVLMs still generate suboptimal captions due to two intrinsic biases: multimodal and linguistic. First, multimodal datasets often contain imbalanced annotations, causing models to over-describe certain objects while glossing over others, leading to inconsistent granularity and reduced caption completeness. Second, inheriting language habits from LLMs (Leng et al., 2024; Liu et al., 2024c), LVLMs tend to favor generic phrasing and frequent co-occurrence patterns, resulting in visual hallucinations: descriptions of non-existent objects or attributes that misrepresent the image. Together, these biases hinder the generation of high-quality, faithful, and fully detailed captions.

To mitigate these limitations, recent efforts (Li et al., 2024; Sun et al., 2024) have explored the use of auxiliary tools or expert modules, such as object detectors (Fang et al., 2023) or image taggers (Zhang et al., 2024) to enrich captions or reduce hallucinations (Yin et al., 2024). While these designs can offer targeted improvements, the overall caption quality is ultimately bound by the precision and coverage of the supporting tools. Given the combinatorial diversity of real-world objects and their attributes, it is unrealistic to rely on handcrafted or category-specific modules as a general solution. These tool-dependent approaches thus fall short in achieving the breadth and adaptability required for generating truly comprehensive and scalable image descriptions.

In contrast to tool-based approaches, we argue that general-purpose LVLMs already possess sufficient perceptual capacity for rich captioning—if guided properly. In particular, we observe that the lack of detail is not necessarily due to insufficient visual understanding but rather stems from suboptimal information extraction during generation. As shown in Figure. 2, when we explicitly ask for more details about an object that is only roughly described in the original caption, the model can provide precise descriptions. Notably, this perceptual capacity is not confined to large models. We find that even compact LVLMs with only 7B parameters can match the descriptive quality of much larger models when equipped with the right prompting. This observation highlights a promising and cost-effective path toward scalable caption generation: leveraging smaller models with proper guidance, rather than relying on brute-force scaling.

Motivated by this insight, we propose ScaleCap, a scalable debiasing strategy that stimulates the model to revisit and refine the caption through a structured, recurrent process. ScaleCap contains two complementary components: heuristic question answering and contrastive sentence rating. The first

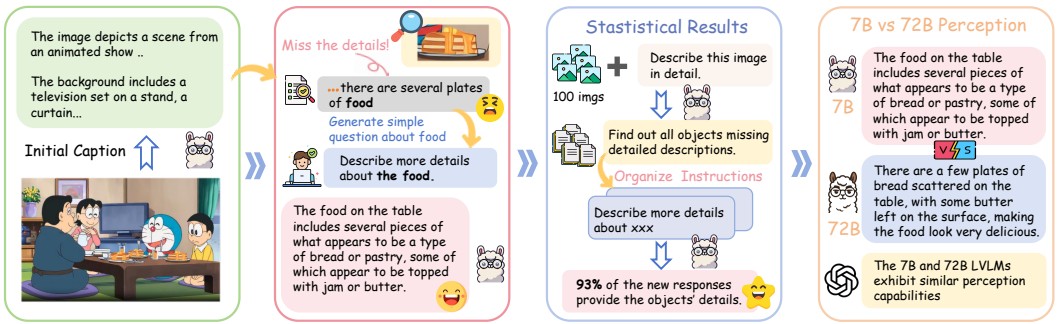

Figure 2: The reason for certain object detail omissions in LVLM captions is mainly due to the absence of guiding heuristic questions rather than insufficient perceptual capability. We also observe that 7B and 72B LVLMs exhibit similar perceptual capabilities.

component prompts a general-purpose LLM to generate content-specific follow-up questions based on an initially generated caption. These questions target under-described or ambiguous elements, such as object attributes or spatial relations, and are then answered by the LVLM to progressively inject additional visual details into the caption. This iterative question-answering loop enables a scalable enrichment process, allowing the model to uncover increasingly fine-grained details with each refinement round. To ensure the factuality and fluency of the evolving caption, both the initial caption and its subsequent refinements are evaluated by the contrastive sentence rating module. This second component addresses hallucinations through an offline sentence-level contrastive decoding strategy, avoiding the coherence issues commonly seen in online decoding. Candidate sentences are generated independently and scored to identify high-quality, visually grounded variants, ensuring that the final caption is not only rich in detail but also coherent and faithful to the image.

We comprehensively evaluate the effectiveness of ScaleCap across three complementary settings. First and most critically, we use ScaleCap to annotate a large-scale dataset of 450K images and apply it to pretrain multiple LVLM architectures. Across all settings, models trained with ScaleCap consistently achieve the best performance on 11 widely used multimodal benchmarks, demonstrating its broad applicability and pretraining benefits. Second, under the Prism framework, we assess caption informativeness via downstream performance and find that ScaleCap-based Qwen2-VL-7B outperforms even larger models like Qwen2-VL-72B—highlighting the efficiency of our strategy. Finally, we evaluate semantic coverage through image reconstruction, showing that ScaleCap captions better preserve visual content than existing open-source models and GPT-4o. These results collectively demonstrate that the captions generated by our approach are highly informative and accurate.

## 2 SCALECAP

In this section, we first introduce the details of the proposed scalable captioning pipeline ScaleCap, then present ScaleCap-450K, a large-scale, high-quality dataset constructed using ScaleCap.

The overall pipeline of ScaleCap is illustrated in Figure 3, which integrates heuristic question answering and contrastive sentence rating in a scalable generation-refinement framework. Specifically, given an input image, the model is prompted to generate an initial caption, and the contrastive sentence rating module is applied to extract high-quality sentences from it. We denote it as the "golden sentences", which is the starting point of the ScaleCap. Building upon these golden sentences, the heuristic question answering module generates a series of content-relevant questions to explore additional visual details. Each answer is then evaluated by the contrastive sentence rating module to filter out hallucinated or low-quality content. As more questions are raised, the caption is progressively enriched with finer-grained and more balanced descriptions. At last, we use a capable LLM to integrate the complex and abundant visual information into a complete and structured image caption. To manage inference overhead, the process is governed by a pre-defined scale budget, which limits the maximum number of questions that can be asked. ScaleCap's scalable refinement strategy flexibly balances caption quality and computational cost, producing informative and faithful outputs.

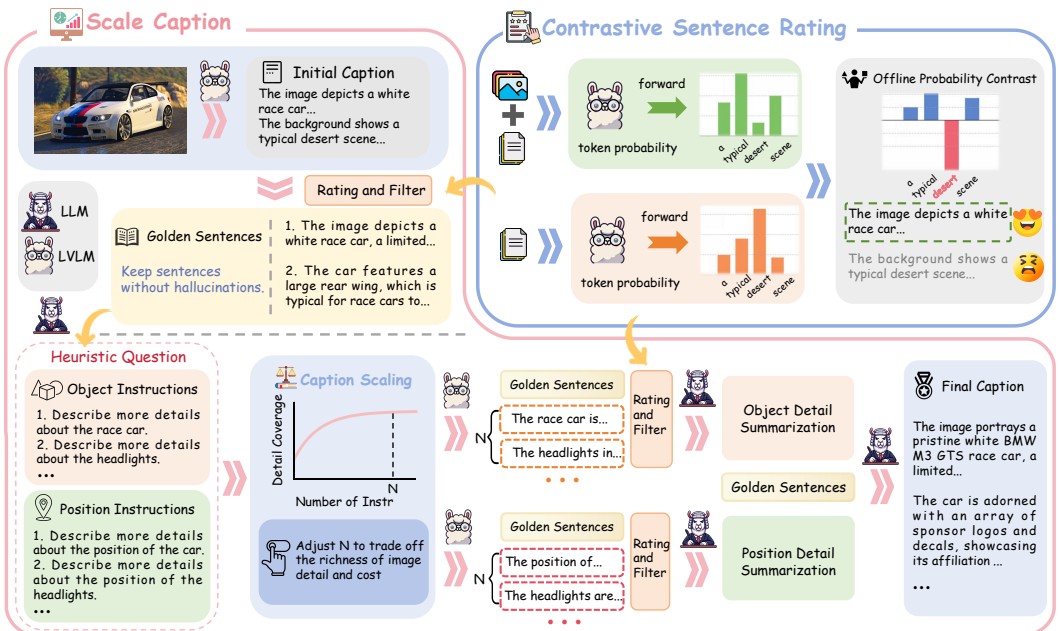

Figure 3: **Overview of ScaleCap.** ScaleCap is composed of two synergistic parts: heuristic question answering and contrastive sentence rating. The first module utilizes a general-purpose LLM to create guiding questions, and the second module addresses hallucinations by an offline contrastive strategy.

## 2.1 HEURISTIC QUESTION ANSWERING MODULE

**Heuristic question raising.** To extract richer object- and position-level details, we generate simple, structured instructions (e.g., "Describe more details about the airplane") based on golden sentences. We use a powerful LLM $\mathcal{M}_L$ in an in-context manner for instruction generation. Let $S_G = \{S_1, S_2, \ldots, S_q\}$ denote the golden sentence set. For each $S_k$, we construct an object prompt $T_{ict}$ with in-context examples to guide the LLM in generating a set of object-related instructions $I_o^k = I_{o,1}^k, I_{o,2}^k, \ldots, I_{o,k_v}^k$, where each $I_{o,i}^k$ queries additional details about an object in $S_k$. Together, $I_o^k$ covers all objects mentioned in $S_k$. This process can be mathematically represented as

$$I_o^k = \mathcal{M}_L(T_{ict}, S_k), \quad k = 1, 2, \ldots, q \tag{1}$$

Likewise, we generate positional instructions to capture the spatial relationships between objects and the overall image layout. Building on the object instruction set $I_o$, we construct the position instruction set $I_p$ by simply adding a position-specific prefix to each object instruction, yielding prompts like "Describe more details about the position of the airplane."

For the full set of golden sentences $S_G = \{S_1, S_2, \ldots, S_q\}$, we generate both object and position instructions per sentence, forming $I_o = \bigcup_{k=1}^q I_o^k$ and $I_p = \bigcup_{k=1}^q I_p^k$. Together, these instruction sets support a comprehensive understanding of object appearances and their position. To manage inference overhead, the process is governed by a pre-defined scale budget N, which limits the maximum number of object and position instructions. In the following experiments, unless otherwise specified, N is generally set to a large value to include all instructions.

**Efficient visual answering.** As discussed in Prism (Qiao et al., 2024), LVLMs exhibit comparable perceptual capabilities across scales, with differences mainly in reasoning. Since the constructed $I_o$ and $I_p$ are straightforward and require minimal reasoning, a small-scale LVLM can effectively handle these instructions. Thus, we directly apply $I_o$ and $I_p$ to a lightweight LVLM to extract fine-grained image details at low cost. Formally, for any instruction $I_{o,i}^k$, the object-specific details are obtained as $D_{o,i}^k = \mathcal{M}_V(I, I_{o,i}^k)$, where both the image $I$ and instruction $I_{o,i}^k$ are input to the model. By processing $I_o$ and $I_p$, we collect object details $D_o$ and position details $D_p$.

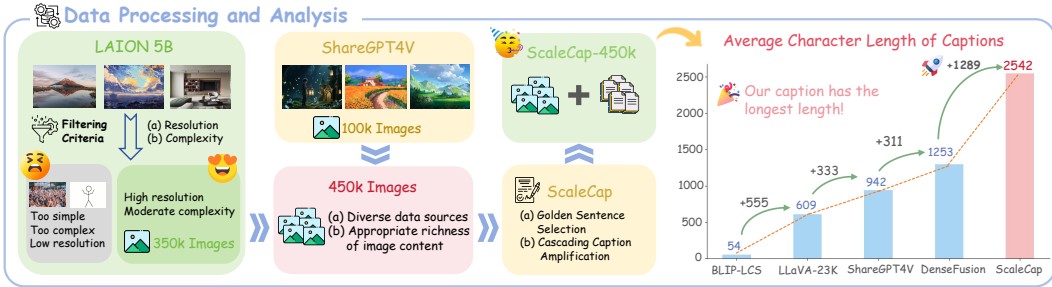

Figure 4: **Data processing and analysis.** During the image collecting and processing stage, we primarily focus on the diversity and richness of image content. In the resulting ScaleCap-450k, the captions are significantly longer than those in other datasets.

## 2.2 CONTRASTIVE SENTENCE RATING MODULE

**Basic Formulation.** Let $\mathcal{M}_V$ denote the LVLM parameterized by $\theta$. Given an image $I$ and a captioning instruction $T$, the model generates a caption $C$ as: $C = \mathcal{M}_V(I, T)$. where $C = \{c_1, c_2, \ldots, c_n\}$ is a sequence of $n$ tokens generated autoregressively. At each decoding step $t$, the token $c_t$ is drawn from the conditional distribution: $y_t \sim p_\theta(y_t \mid I, [T, c_{<t}])$, with $y_t$ denoting the predicted token and $c_{<t}$ the preceding context. The probability distribution is conditioned on both the image $I$ and the textual input $[T, c_{<t}]$, guiding the model to iteratively generate each token.

**Offline Contrastive Probability Analysis.** To detect hallucinated content without disrupting the natural language distribution, we adopt an offline contrastive probability analysis, in contrast to prior online decoding methods (Leng et al., 2024; Wang et al., 2024; Huo et al., 2024). Given the initial caption tokens $C = \{c_1, c_2, \ldots, c_n\}$, we compute two sequences of token probabilities: caption tokens with and without the image input. The probability sequence $P$ conditioned on the image $I$ is:

$$P = \{p_1, p_2, \ldots, p_n\}, \qquad p_t = p_\theta(y_t = c_t \mid I, [T, c_{<t}]), \quad t \in [1, n]. \tag{2}$$

Here, $p_t$ denotes the likelihood of generating token $c_t$ given the image $I$, the instruction prompt $T$, and previous tokens $c_{<t}$. We then compute the counterpart $P'$ by conditioning only on textual input:

$$P' = \{p'_1, p'_2, \ldots, p'_n\}, \qquad p'_t = p_\theta(y_t = c_t \mid [T, c_{<t}]), \quad t \in [1, n]. \tag{3}$$

This sequence reflects the model's inherent linguistic prior—how likely it is to generate $c_t$ without visual evidence. Since large vision-language models (LVLMs) inherit strong language modeling capabilities from LLMs, they may over-rely on textual co-occurrence patterns, leading to hallucinated outputs (Leng et al., 2024; Liu et al., 2024c). To quantify this effect, we define the contrastive probability sequence $\Delta P$ as:

$$\Delta P = P - P' = \{\Delta p_1, \Delta p_2, \ldots, \Delta p_n\}, \qquad \Delta p_k = p_k - p'_k. \tag{4}$$

A high $\Delta p_k$ indicates that token $c_k$ benefits significantly from the visual context and is thus more likely grounded in the image. In contrast, a low $\Delta p_k$ suggests that the token is generated primarily based on language priors, signaling potential hallucination.

**Sentence-Level Rating.** To mitigate hallucinations without disrupting fluency, we filter at the sentence level rather than at the token level. The initial caption $C$ is segmented into sentences as $C = \{C_1, C_2, \ldots, C_m\}$ using punctuation (e.g., periods), where $m$ is the number of sentences. Sentence $C_k = \{c_1^k, c_2^k, \ldots, c_{k_l}^k\}$ consists of $k_l$ tokens, and each token associates with a contrastive probability difference $\Delta p_i^k$ from Equation 4, forming a sentence-wise sequence $\Delta P_k = \{\Delta p_1^k, \ldots, \Delta p_{k_l}^k\}$.

To assess whether a sentence is language-biased, we compute the maximum $\Delta p_i^k$ over critical tokens[1]. Sentences with strong visual grounding are retained as the Golden Sentences $S_G$:

$$S_G = \{C_k \mid \max\{\Delta p_1^k, \Delta p_2^k, \ldots, \Delta p_{k_l}^k\} > \tau\} \tag{5}$$

where $\tau$ is a tunable threshold, with higher values leading to stricter filtering.

---

[1]Identified via part-of-speech tagging to exclude function words such as adpositions.

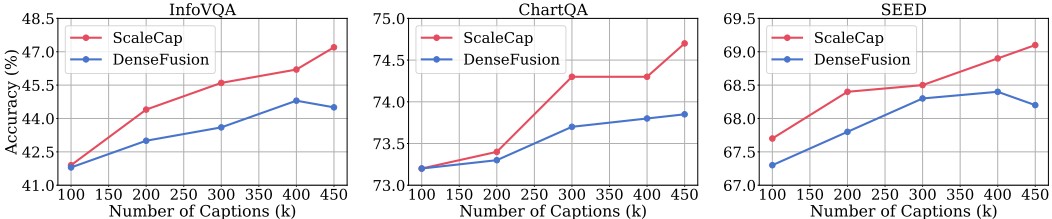

Figure 5: The scaling trend under different number of pretraining data.

## 2.3 CAPTION INTEGRATION

Our ultimate goal is to form a complete and structured image caption. Leveraging the strong summarization and logical reasoning capabilities of the LLM, we instruct it to organize and consolidate the relatively fragmented content from the detail sets using two prompts, $T_o$ and $T_p$, for object-level details and position-level details respectively. To ensure coherence and structure during summarization, we also provide the LLM with the golden sentences as a caption backbone. This helps the LLM maintain awareness of the overall caption structure, enhancing the quality of the summarization. As a result, we generate the following summaries:

$$C_o = \mathcal{M}_L(S_G, T_o, D_o), \quad C_p = \mathcal{M}_L(S_G, T_p, D_p), \tag{6}$$

where $C_o$ summarizes the object-level details, and $C_p$ summarizes the positional details of the objects in the image. Finally, we utilize the LLM once more to integrate $C_o$ and $C_p$ into a comprehensive final caption. The final caption is generated as follows:

$$F_c = \mathcal{M}_L(S_G, T_{final}, C_o, C_p), \tag{7}$$

where $T_{final}$ is the integration prompt. All prompts used in ScaleCap can be found in Appendix.D.

## 2.4 SCALECAP-450K DATASET

Based on ScaleCap, we create a hyper-detailed image caption dataset as Figure. 4 presents. We collect 450,000 images and then annotate them using ScaleCap to generate high-quality image-text pairs.

**Data Source and Processing.** In collecting images for our dataset, we primarily focus on two aspects: diversity and richness of image content. Given that the ShareGPT4V-100k already includes a wide range of categories, such as artworks, landmarks, etc., it inherently offers a certain level of diversity. Therefore, we opt to directly incorporate these images into our dataset. To further enhance the dataset's diversity and to obtain more content-rich images, we additionally select 350k images from the LAION-5B (LAION, 2022) dataset which encompass a wide range of subjects, styles due to its web-based origin. During filtering, we retain only images with high resolution and moderate complexity. The specific tools and details used in filtering can be found in the Appendix.F.

**Caption Model Selection.** As previously mentioned, ScaleCap leverages the collaboration between a Vision Language Model and a Large Language Model to generate high-quality captions. For LVLM, as we discussed above, a small model is capable of capturing visual content, so we use Qwen2-VL-7B by default. When it comes to the LLM, the question-raising task is relatively simple, while the integration of complex and abundant visual information within thousands of tokens requires advanced reasoning capability, so we resort to Qwen2-72B based on an empirical study.

## 3 PRETRAINING EXPERIMENTS

To comprehensively evaluate the effectiveness of the ScaleCap-450k dataset, we conduct extensive pretraining experiments. The experimental details are as follows.

## 3.1 IMPLEMENTATION DETAILS

Our model structure follows LLaVA-NeXT (Liu et al., 2024b), comprising a vision encoder, an MLP projector, and an LLM. To thoroughly evaluate our approach, we experiment with three configurations: (1) Qwen2.5-7B + Qwen2.5-ViT, (2) Qwen2.5-3B + Qwen2.5-ViT, and (3) InternLM2.5-7B + CLIP-ViT-L/14-336. Training involves three stages: initial pretraining on BLIP-558K, further pretraining

Table 1: **Comparison with different datasets on 11 benchmarks.** ScaleCap-450k significantly improves pretraining efficiency, achieving the best results on nearly all benchmarks with the same amount of data. **Further stringent ablations about image sources** can be found in Appendix B.

| Model | Pretraining Data | Info VQA | Doc VQA | Chart QA | MM Star | Math Vista | LLaVA Bench | MMVet | MMB | MMMU | SEED | AI2D | Average |
|---|---|---|---|---|---|---|---|---|---|---|---|---|---|
| Qwen2.5-7B + Qwen2.5-ViT | Vanilla | 46.2 | 81.5 | 75.5 | 47.0 | 47.0 | 72.8 | 46.7 | 74.6 | 45.2 | 69.9 | 71.6 | 61.6 |
| | ShareGPT4V-450k | 47.5 | 82.9 | 76.0 | 48.8 | 46.2 | 72.9 | 48.9 | 75.2 | 43.8 | 71.3 | 72.7 | 62.4 |
| | DenseFusion-450k | 49.4 | 84.8 | 77.1 | **49.2** | 47.5 | 70.3 | 52.4 | 73.1 | 44.3 | 70.6 | 73.9 | 63.0 |
| | ScaleCap-450k | **51.8** | **85.7** | **77.8** | 48.8 | **49.7** | 74.7 | **55.9** | 75.6 | 46.1 | 71.6 | 74.0 | **64.7** |
| Qwen2.5-3B + Qwen2.5-ViT | Vanilla | 39.1 | 76.3 | 72.1 | 44.8 | 41.6 | 66.4 | 39.9 | 69.1 | 37.4 | 67.1 | 69.7 | 56.7 |
| | ShareGPT4V-450k | 42.4 | 78.3 | 73.0 | 44.8 | 43.7 | 66.8 | 43.3 | 69.5 | 38.5 | 67.9 | 70.2 | 58.0 |
| | DenseFusion-450k | 44.5 | 81.1 | 73.8 | 43.5 | 45.9 | **69.4** | 39.9 | 68.7 | **42.1** | 68.2 | 70.8 | 58.9 |
| | ScaleCap-450k | **47.2** | **81.7** | **74.7** | **44.9** | **46.1** | 68.3 | **45.6** | **70.0** | 41.3 | **69.1** | **71.8** | **60.1** |
| InternLM2.5-7B + CLIP-ViT-L | Vanilla | 36.2 | 72.3 | 68.4 | 47.8 | 43.5 | 68.7 | 41.3 | 72.2 | 41.2 | 72.7 | 73.9 | 58.0 |
| | ShareGPT4V-450k | 36.9 | 72.3 | 69.1 | 48.6 | 42.3 | 66.6 | 45.8 | 72.5 | 41.5 | 73.2 | 72.6 | 58.3 |
| | DenseFusion-450k | 39.1 | 75.1 | 70.0 | 48.7 | 44.1 | 66.9 | 47.2 | 72.2 | 40.1 | 73.6 | 73.0 | 59.1 |
| | ScaleCap-450k | **39.6** | **75.5** | **71.3** | **48.7** | **44.5** | 70.3 | **48.0** | **73.4** | **42.1** | **74.0** | **74.7** | **60.2** |

Table 2: Caption informativeness comparison results in Prism Framework. ScaleCap significantly outperforms the quality of captions generated by Qwen2-VL-7B and Qwen2-VL-72B.

| Caption Strategy | LVLM | MM Vet | MM Star | Info VQA | Chart QA | Text VQA | Average |
|---|---|---|---|---|---|---|---|
| Prism | Qwen2-VL-7B | 53.3 | 47.7 | 49.3 | 68.5 | 51.7 | 54.1 |
| Prism | Qwen2-VL-72B | 57.3 | 48.7 | 50.0 | 69.5 | 54.4 | 56.0 |
| ScalCap | Qwen2-VL-7B | 58.8 | 50.3 | 53.8 | 72.9 | 55.3 | 58.2 |

Table 3: Ablation study of Object Instructions and Position Instructions on benchmarks subset.

| Method | Text VQA | MM Vet | Chart QA | Avg |
|---|---|---|---|---|
| Only Object Instr | 52.9 | 54.5 | 69.1 | 58.8 |
| Only Position Instr | 52.3 | 54.3 | 65.7 | 57.4 |
| ScaleCap | 53.2 | 58.8 | 72.5 | 61.5 |

using high-quality image caption, and final instruction-tuning with Open-LLaVA-NeXT-Instruct-1M (Chen & Xing, 2024). Baselines for comparison include a two-stage Vanilla method (without further pretraining), ShareGPT4V-450k (Chen et al., 2023), and DenseFusion-450k (Li et al., 2024). To ensure fair evaluation, differences are limited to the further pretraining datasets only. Details of the settings can be found in the Appendix.G.

### 3.1.1 MAIN RESULTS

**Comparison with different pretraining datasets.** As shown in Table 1, comprehensive experiments under three settings demonstrate that pretraining with ScaleCap-450k consistently yields the best performance across most benchmarks. For instance, in Qwen2.5-7B setting, ScaleCap-450k improved InfoVQA scores by 4.3% over ShareGPT4V-450k and 2.4% over DenseFusion-450k. On natural image QA benchmarks like MMVet, ScaleCap-450k achieved a 7% gain over ShareGPT4V-450k and 3.5% over DenseFusion. Similar gains were observed with other LLMs.

Compared to DenseFusion, which uses captions from multiple expert models that often miss object details due to limited attribute coverage, ScaleCap leverages general-purpose models for higher-quality captions. These detailed descriptions of objects significantly enhance modality alignment during pretraining. Consequently, the better-aligned visual features are more readily understood by the LLM, enabling finer-grained image comprehension and improved benchmark performance.

**The pre-training data scaling performance.** We then study the data efficiency of captions generated by ScaleCap. We conduct pretraining using varying data volume from 100K to 450K, with the Qwen2.5-3B setting. The results show that, given the same pretraining samples, modality alignment using the ScaleCap dataset significantly outperforms the DenseFusion dataset. Moreover, as the pretraining data volume increases, the advantage of ScaleCap becomes even more pronounced. These findings indicate that captions generated by ScaleCap enable more efficient modality alignment. Additionally, the steep upward trend of the ScaleCap curve suggests that further expanding the data volume will continue to yield substantial performance gains.

Table 4: ScaleCap, equipped with GPT-4o in the Prism framework and supplemented with image-visible responses, outperforms other advanced proprietary models.

| Method | Sonnet3.5 | GPT4V | GPT4o | Gemini-2.0-Pro | Qwen2-VL-72B | ScaleCap (GPT4o+GPT4o) |
|---|---|---|---|---|---|---|
| MMVet | 66.0 | 67.5 | 69.1 | 70.4 | 74.0 | 76.1 |

Table 5: Ablation study on the summarization model scale in ScaleCap on benchmarks subset.

| Summarization Model | MMVet | MMStar |
|---|---|---|
| Qwen2-7B | 43.6 | 40.3 |
| Qwen2-72B | 58.8 | 49.5 |

## 4 DIVE INTO SCALECAP

### 4.1 VALIDATING INFORMATIVENESS OF SCALECAP VIA VQA

**Setup.** Prism (Qiao et al., 2024) is a framework designed to disentangle the perception and reasoning processes of LVLMs. It separates the problem-solving pipeline into two stages: in the perception stage, LVLMs extract information from images and convert it into textual form without access to the question, thus producing general-purpose captions; in the reasoning stage, LLMs generate answers based on the extracted text. With a fixed LLM, the benchmark performance directly reflects the informativeness contained within the caption. We use Qwen2-72B as the LLM in the Prism framework to answer visual questions based on captions. For comparison, we provide the carefully designed Generic Instruction used in (Qiao et al., 2024) as a prompt to Qwen2-VL-7B and Qwen2-VL-72B to generate as detailed a caption as possible, establishing strong baselines.

**Main Results.** As illustrated in Table 2, we can observe that ScaleCap outperforms both Qwen2-VL-7B and Qwen2-VL-72B across all benchmarks, demonstrating outstanding performance. These results strongly validate the informativeness of caption generated by ScaleCap, demonstrating that its heuristic question answering effectively extracts fine-grained image details. This significantly enhances the caption quality initially generated by Qwen2-VL-7B, even surpassing Qwen2-VL-72B by a substantial margin. These findings indicate that ScaleCap can generate captions of significantly higher quality than those produced by a large LVLM trained on extensive image-text data.

**The Potentials of ScaleCap.** ScaleCap is a flexible captioning pipeline that allows for the replacement of both the LVLM and LLM with any open-source or proprietary models. This flexibility suggests significant potential, especially when equipped with powerful models like GPT-4o, which could enhance caption quality. To test this, we compare ScaleCap with GPT-4o against other advanced models, including Sonnet 3.5 and Gemini-2.0-Pro. In experiments, ScaleCap generates the question-irrelevant captions first, with the LVLM's direct image-visible response also attached as supplemental information to the LLM. As shown in Table 4, ScaleCap not only outperforms its direct answering baseline but also all proprietary LVLMs, indicating its high potential.

### 4.2 VALIDATING INFORMATIVENESS OF SCALECAP VIA RECONSTRUCTION

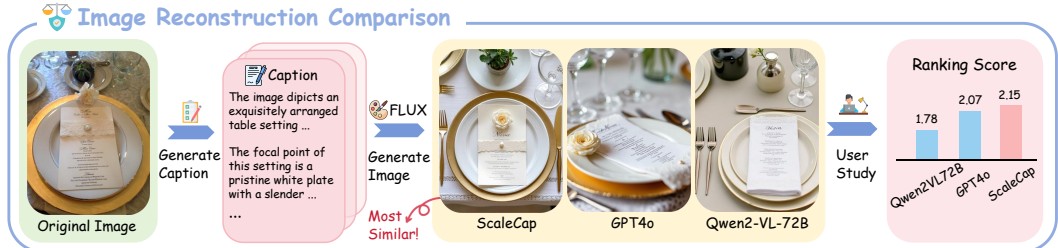

Figure 6: Human evaluation of image similarity with original image over 50 samples and 25 volunteers. Volunteers rank the images based on their similarity to the original image.

Leveraging the capabilities of powerful modern text-to-image generation models, the similarity between a reconstructed image and its original can effectively reflect the extent to which a caption covers the image's content. To gain an intuitive understanding of caption quality, we utilize one of the best image generation models, FLUX. FLUX can adhere to very detailed instructions, such as "wearing a silver cross-shaped necklace around his neck." As a result, it is an ideal tool for validating caption quality. When the caption is highly detailed and covers every object in the image, the FLUX-generated image shows a high degree of similarity to the original image. Conversely, if the caption contains errors or lacks detail, the similarity will be much lower. We randomly sampled 50 images and used ScaleCap, GPT4o, and Qwen2-VL-72B to generate captions. FLUX then generate

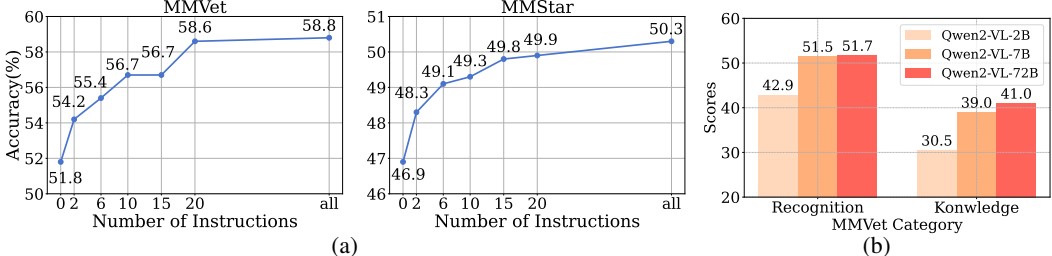

Figure 7: (a) Caption scaling. We adjust the number of instructions N used in ScaleCap to explore changes in benchmark performance within the Prism framework. (b) As the LVLM in ScaleCap scales up, perception capability saturates at 7B, while world knowledge continues to increase.

corresponding images based on these captions. Finally, we invite 25 volunteers to rank the similarity of the generated images to the originals. A model that ranks first in the three categories receives three points. As shown in Figure. 6, captions generated by ScaleCap result in images that more accurately reflect the original images than those generated by GPT4o, significantly surpassing Qwen2-VL-72B. This further demonstrates the superior quality of our approach.

## 4.3 ANALYSIS ON SCALECAP COMPONENTS

**Caption quality increases with scaling of inference budgets N.** In ScaleCap, we adjust the number of heuristic questions to trade off between the richness of image detail and computational cost. To intuitively present this effect through the Prism framework, we conduct experiments on MMVet and MMStar. The results in Figure 7 show that initially increasing the number of heuristic questions leads to a sharp rise in benchmark performance, indicating that these questions significantly enrich image detail. However, when the number exceeds 20, the performance curve begins to plateau, suggesting that most objects mentioned in VQA tasks are already covered. This trend aligns well with the general scaling laws observed in previous work.

**LVLM with 7B size is sufficient and efficient.** In this section, we study the scale of the LVLM used for visual information extraction. Here we use the Qwen2-VL series from 2B to 72B and evaluate the benchmark performance under the Prism setting above. Due to the high inference cost of large LVLMs, we use a randomly sampled 300-question subset from MMStar, ChartQA, and TextVQA. As shown in the first part of Table 6, we find that LVLM from 2B to 7B shows a promising improvement, but the further improvement for 72B is minor. This is consistent with the conclusion in Prism and proves our analysis that a 7B size model is capable enough for extracting most visual information.

**Large LVLM introduces more world knowledge in the caption.** Despite the minor gap between 7B and 72B models, we further analyze the difference in Figure 7b by different categories of questions in MMVet. We find the models perform similarly at the recognition question, but larger models get better performance on questions that are related to world knowledge, which is reasonable because a model with larger size contains more knowledge and could introduce it in the caption.

**Small LLM struggles with complex visual information integration.** Then we study the influence of the LLM used for heuristic question raising and information integration. As shown in the second part of Table 6, the 7B LLM shows significantly worse performance than the 72B model. To position the performance bottleneck, we first compare the question-raising quality and find the gap is minor. So we use the same context before the Caption Integration Phase and evaluate the integration quality between models in Table 5, and observe a consistent performance gap with Table 6. This indicates a substantial difference in summarization capabilities among model sizes. During summarization, the combined object details can result in a context length of up to 20k tokens, which causes Qwen2-7B to miss important information due to its limited ability to handle long contexts.

**Object and Position Instructions are equally important.** In ScaleCap, questions are raised for both the object and its position. Here we study their effectiveness in Table 3 within the Prism framework. Removing either Object Instructions or Position Instructions results in a notable drop in performance.

**Ablating Contrastive Sentence Rating Strategy.** Here we study the effectiveness of the Contrastive Sentence Rating Module on CHAIR, a benchmark specialized for hallucination evaluation. As

Table 6: Ablation on LVLM and LLM scale in benchmarks subset. As the LVLM scales up, caption quality saturates from 7B; as the LLM scales up, caption quality continues to improve.

| LVLM | LLM | MM Vet | MM Star | Chart QA | Text VQA | Average |
|------|-----|--------|---------|----------|----------|---------|
| Qwen2-VL-2B | Qwen2-72B | 54.0 | 43.6 | 58.8 | 44.4 | 50.2 |
| Qwen2-VL-7B | Qwen2-72B | 58.8 | 49.5 | 72.5 | 53.2 | 58.5 |
| Qwen2-VL-72B | Qwen2-72B | 59.0 | 52.3 | 69.4 | 54.1 | 58.7 |
| Qwen2-VL-7B | Qwen2-7B | 45.5 | 40.0 | 53.0 | 47.5 | 46.5 |
| Qwen2-VL-7B | Qwen2-72B | 58.8 | 49.5 | 72.5 | 53.2 | 58.5 |

Table 7: Hallucination evaluation results. Golden Sentence Selection strategy performs the best.

| Method | $\text{CHAIR}_S\downarrow$ | $\text{CHAIR}_I\downarrow$ |
|--------|--------|--------|
| LLaVA-v1.5 7B | 48.8 | 13.9 |
| +VCD | 46.8 | 13.2 |
| +OPERA | 44.6 | 12.8 |
| +Golden Sentence | 33.6 | 11.3 |
| Qwen2-VL 7B | 44.2 | 7.5 |
| +Golden Sentence | 25.8 | 6.8 |

shown in Table 7, our strategy significantly mitigates hallucination in LLaVA1.5, outperforming baselines such as OPERA (Huang et al., 2024) and VCD (Leng et al., 2024). It also achieves notable hallucination reduction on capable models like Qwen2-VL-7B, proving the effectiveness and generalization of the Contrastive Sentence Rating Module in hallucination elimination.

## 5 CONCLUSION

In this work, we present ScaleCap, a scalable image captioning framework. By integrating heuristic question answering and contrastive sentence rating, ScaleCap progressively enriches and calibrates captions with increased inference budget, resulting in more detailed and balanced descriptions. Extensive experiments demonstrate that captions generated by ScaleCap not only excel in downstream tasks such as VQA and image reconstruction, but also significantly boost LVLM pretraining when used at scale, paving the way for high-quality captioning systems.

## ACKNOWLEDGMENTS

This project is funded in part by Shanghai Artificial Intelligence Laboratory, Shanghai Innovation Institute, the Centre for Perceptual and Interactive Intelligence (CPII) Ltd under the Innovation and Technology Commission (ITC)'s InnoHK. Dahua Lin is a PI of CPII under the InnoHK.

## ETHICS STATEMENT

This work builds on large vision-language models (LVLMs) to develop scalable and debiased image captioning. Our approach primarily uses publicly available datasets such as LAION-5B and ShareGPT4V, which are filtered to ensure appropriate resolution, diversity, and complexity. No personally identifiable, sensitive, or private data were intentionally collected. We acknowledge that large-scale image-text datasets may contain biases or mislabeled content, which can propagate into downstream models. To mitigate these risks, our method explicitly targets multimodal and linguistic biases by filtering hallucinations and promoting balanced detai. In addition, the image data undergo rigorous cleaning to further enhance reliability. The generated captions are intended to support research in multimodal alignment and are not designed for high-stakes decision-making scenarios such as medical or legal applications. We emphasize that any deployment of this work should carefully consider fairness, potential misuse, and compliance with privacy and copyright regulations. This research adheres to the ICLR Code of Ethics.

## REPRODUCIBILITY STATEMENT

We have made significant efforts to ensure the reproducibility of our work. The detailed architecture of ScaleCap, including heuristic question answering and contrastive sentence rating, is described in Section 2, with all prompts provided in Appendix D. Implementation details of pretraining experiments, including model configurations, training stages, and hyperparameters, are reported in Section 3.1 and Appendix G. The construction of the ScaleCap-450K dataset, including data sources, filtering criteria, and caption generation process, is documented in Section 2.4 and Appendix F. Ablation studies and benchmark comparisons further validate the robustness of our findings. Together, these resources provide sufficient transparency for independent verification and extension of our results.

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

## A  RELATED WORK

**Hallucination in LVLMs.** Hallucination remains a significant challenge in LVLMs (Bai et al., 2024; Cui et al., 2023; Liu et al., 2024a), despite the rapid development of multimodal. Object hallucination occurs when large vision-language models produce textual descriptions that mention objects or attributes that are not actually present in the corresponding image. This issue is commonly seen in tasks like image captioning and visual question answering, where it is essential to ensure a precise correlation between the visual and textual elements (Guan et al., 2024; Li et al., 2023). Currently, a range of methods have been proposed to address hallucination. These methods can be broadly categorized into two types: one requires training, with numerous approaches utilizing diversified training data to enhance the instruction tuning phase (Yu et al., 2024a), while others leverage preference data through DPO or other reinforcement learning strategies to mitigate hallucinations (Sun et al., 2023; Yu et al., 2024c; Zhu et al., 2024). Another approach is training-free (Huo et al., 2024; Wang et al., 2024; Wan et al., 2024; Kan et al., 2024), with notable works such as OPERA (Huang

Table 8: **Ablations about image sources.** We apply ScaleCap to annotate the images in DenseFusion-450k, yielding the ScaleCap-DenseFusion-450k dataset for stringent comparison.

| Model | Pretraining Data | Info VQA | Doc VQA | Chart QA | MM Star | Math Vista | LLaVA Bench | MM Vet | MMB | MMMU | SEED | AI2D | Avg |
|---|---|---|---|---|---|---|---|---|---|---|---|---|---|
| Qwen2.5-3B + Qwen2.5-ViT | Vanilla | 39.1 | 76.3 | 72.1 | 44.8 | 41.6 | 66.4 | 39.9 | 69.1 | 37.4 | 67.1 | 69.7 | 56.7 |
| | DenseFusion-450k | 44.5 | 81.1 | 73.8 | 43.5 | 45.9 | 69.4 | 39.9 | 68.7 | **42.1** | 68.2 | 70.8 | 58.9 |
| | ScaleCap-DenseFusion-450k | **46.7** | **81.7** | **74.9** | **44.1** | **46.0** | **73.8** | **43.9** | **69.9** | 40.8 | **69.2** | **71.2** | **60.2** |

et al., 2024), which employs a novel MLLM decoding method based on an over-trust penalty and a retrospection-allocation strategy that addresses the internal causes of hallucinations. VCD (Leng et al., 2024) contrasts the output distributions derived from original and distorted visual inputs to mitigate over-reliance on statistical biases and unimodal priors. These methods work during the decoding phase, which is online, and detect hallucinations at the token level. We propose a sentence-level hallucination detection method, which is performed after the entire sentence is generated. This approach can enhance the coherence of the sentence and improve the stability of detection.

**Image Caption.** To enhance LVLMs, early works focused on large-scale image-text datasets. CC3M Sharma et al. (2018) and CC12M Changpinyo et al. (2021) leveraged web-crawled alt-text but lacked fine-grained details, while manually annotated datasets like SBU Ushiku et al. (2015) and COCO-Captions Chen et al. (2015) offered higher quality but struggled with contextual richness. Later efforts Fan et al. (2023); Lai et al. (2024); Yu et al. (2024b); Rasheed et al. (2024); Garg et al. (2024); Onoe et al. (2024) improved captions using LLMs. LLaVA Liu et al. (2023) introduced human-annotated captions and bounding boxes to guide GPT-4 but remained annotation-heavy. ShareGPT4V Chen et al. (2024) constructs a large-scale dataset with 1.2M highly descriptive captions, demonstrating significant improvements in LVLMs' performance across multiple benchmarks. DCE Sun et al. (2024) enhances captions with fine-grained attributes and 3D spatial relationships using open-source visual specialists, optimizing cost-effective annotation for complex scenes. Perceptual Fusion Li et al. (2024) leverages high-resolution image processing and multi-expert signal fusion (detection, OCR, tagging) to train a scalable caption engine for open-domain visual perception. There is also work that improves caption quality and mitigates hallucinations by building specialized pipelines. For example, VFC(Ge et al., 2024) integrates an open-vocabulary detector together with multiple LVLMs to synthesize high-quality captions, while CapMAS(Lee et al., 2024) uses an LLM to break the initial caption into independent sentences and then has an LVLM verify them. These methods rely on LVLMs or dedicated models as verifiers, which helps reduce hallucinations to some extent. However, they provide very limited improvement in visual information, and in some cases may even reduce it. In contrast, our dual-modality debiasing strategy achieves substantial gains in both aspects.

## B ABLATIONS ABOUT IMAGE SOURCES IN PRETRAINING EXPERIMENTS

In our previous experiments Table 1 , we did not strictly control for ScaleCap and other datasets to use exactly the same set of images. As a result, factors such as image resolution and type could potentially influence the comparison. To mitigate this concern, we conducted an additional ablation study on image sources under the Qwen2.5-3B + Qwen2.5-ViT setting. Specifically, we applied ScaleCap to annotate the images in DenseFusion-450k, yielding the ScaleCap-DenseFusion-450k dataset. The corresponding experimental results are presented in Table 8. The results show that, under the condition of using exactly the same set of images, pretraining with ScaleCap-DenseFusion-450k yields an average improvement of 1.3% over DenseFusion-450k, and outperforms the baseline on 10 benchmarks. This stringent comparison demonstrates that the superiority of the ScaleCap-450k dataset does not stem from image diversity or resolution, but rather from the exceptionally high quality of the captions generated by ScaleCap itself.

## C LLM SIZE EFFECT ON QUESTION GENERATION

We explicitly evaluated the robustness of our question-generation component to both LLM size and domain shift. Using Qwen2VL-7B to produce initial captions for 500 natural images, we then generated questions with Qwen2-7B-Instruct and Qwen2-72B-Instruct under the same ScaleCap prompt. The accuracy of the generated questions is verified through a strict format matching process. The result is in Table 9. Across domains, the question-format accuracy remained consistently

---

**Algorithm 1** ScaleCap: Scalable Image Captioning via Dual-Modality Debiasing

---

**Require:** Image $I$, LVLM $\mathcal{M}_V$, LLM $\mathcal{M}_L$, Scale Budget $N$, Threshold $\tau$
**Require:** Prompts: $T_{init}, T_{ict}, T_{final}$
**Ensure:** Final Caption $C_{final}$

    **Stage 1: Initial Generation & Debiasing**
1:  $C_{init} \leftarrow \mathcal{M}_V(I, T_{init})$
2:  $S_G \leftarrow \text{CONTRASTIVESENTENCERATING}(C_{init}, I, \tau)$

    **Stage 2: Heuristic Question Answering**
3:  $I_o, I_p \leftarrow \emptyset$
4:  **for** each sentence $S_k \in S_G$ **do**
5:     $I_q^k \leftarrow \mathcal{M}_L(T_{ict}, S_k)$
6:     $I_p^k \leftarrow \text{ADDPOSITIONPREFIX}(I_o^k)$
7:     $I_o \leftarrow I_o \cup I_o^k, \quad I_p \leftarrow I_p \cup I_p^k$
8:  **end for**
9:  $D_o, D_p \leftarrow \emptyset$
10: $Q_{queue} \leftarrow \text{SELECT}(I_o, N) \cup \text{SELECT}(I_p, N)$
11: **for** each question $q \in Q_{queue}$ **do**
12:     $A_{raw} \leftarrow \mathcal{M}_V(I, q)$
13:     $A_{clean} \leftarrow \text{CONTRASTIVESENTENCERATING}(A_{raw}, I, \tau)$
14:     **if** $q \in I_o$ **then**
15:         $D_o \leftarrow D_o \cup A_{clean}$
16:     **else**
17:         $D_p \leftarrow D_p \cup A_{clean}$
18:     **end if**
19: **end for**

    **Stage 3: Caption Integration**
20: $C_o \leftarrow \mathcal{M}_L(S_G, T_o, D_o)$
21: $C_p \leftarrow \mathcal{M}_L(S_G, T_p, D_p)$
22: $C_{final} \leftarrow \mathcal{M}_L(S_G, T_{final}, C_o, C_p)$
23: **return** $C_{final}$

---

24: **procedure** CONTRASTIVESENTENCERATING$(C_{raw}, I, \tau)$
25:     Split $C_{raw}$ into sentences $\{s_1, s_2, ..., s_m\}$
26:     $S_{filtered} \leftarrow \emptyset$
27:     **for** each sentence $s_j$ in $C_{raw}$ **do**
28:         Compute probs with image: $P = p_\theta(y|I, [T, c_{<t}])$
29:         Compute probs w/o image: $P' = p_\theta(y|[T, c_{<t}])$
30:         Compute difference: $\Delta P = P - P'$
31:         **if** $\max(\Delta P) > \tau$ **then**
32:             $S_{filtered} \leftarrow S_{filtered} \cup \{s_j\}$
33:         **end if**
34:     **end for**
35:     **return** $S_{filtered}$
36: **end procedure**

---

high: for natural images, Qwen2-7B and Qwen2-72B achieved 97.5% and 98.7%; for Qwen2-72B on Infographic and STEM images (additional 500 images), accuracy reached 98.8% and 97.7%, respectively. This suggests that question generation is a relatively simple, robust task.

## D   PROMPTS USED IN SCALECAP

Prompts used in ScaleCap and Prism are presented in Figure 8 and Figure 9.

Table 9: LLM size effect on question generation

| Instruction Generation Model | Domain | Instruction Accuracy |
|---|---|---|
| Qwen2-7B | natural | 97.5% |
| Qwen2-72B | natural | 98.7% |
| Qwen2-72B | Infografic | 98.8% |
| Qwen2-72B | STEM | 97.7% |

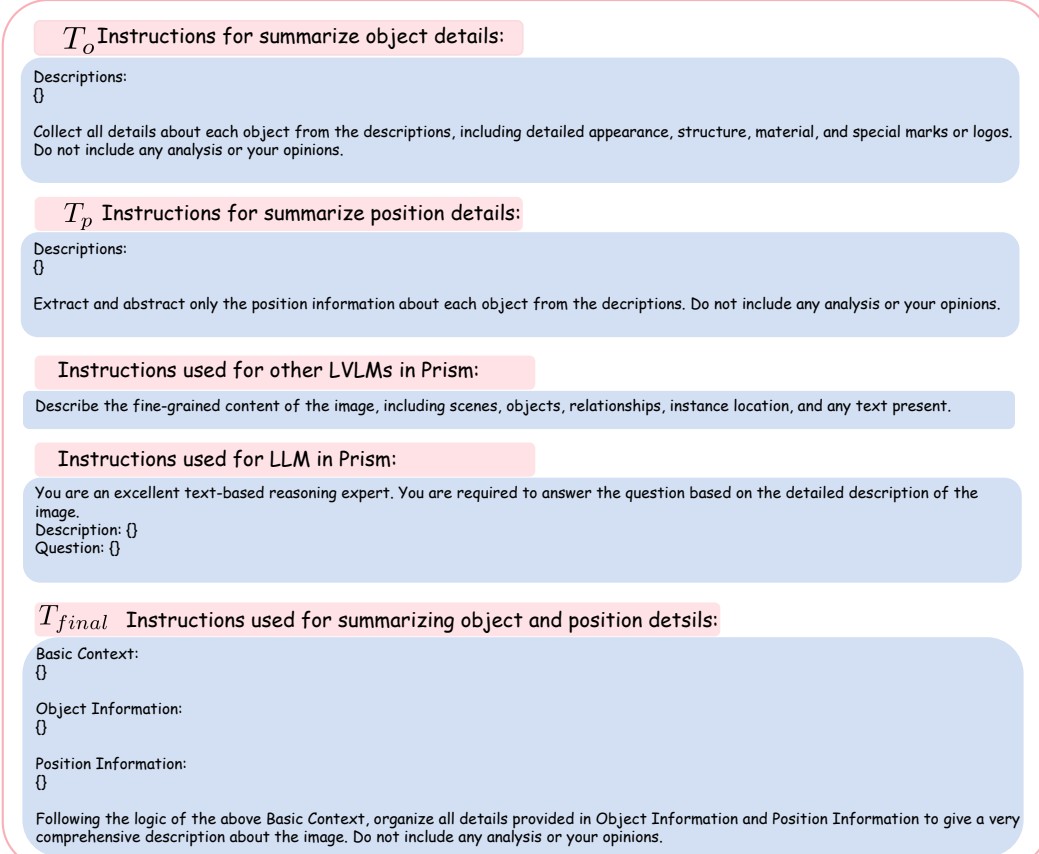

Figure 8: Prompts used in ScaleCap and Prism.

# E   LARGE LANGUAGE MODEL USAGE

During the preparation of this paper, large language models (LLMs) were used exclusively as tools for language refinement and did not contribute to the methodological or analytical aspects of the study. Their main function was to support the writing process by improving grammar, adjusting phrasing, and harmonizing style across different sections of the manuscript. This assistance helped us present complex technical details with greater clarity and coherence, ensuring that our arguments and findings were communicated effectively to the reader. In addition, LLMs were employed to identify and correct minor linguistic issues or awkward sentence structures that could otherwise obscure the scientific content. We emphasize that their involvement was strictly limited to these editorial tasks. The design of the research framework, the execution of experiments, the analysis of results, and the formulation of conclusions were carried out entirely by the authors. By restricting the role of LLMs to editing and proofreading, we were able to take advantage of their strengths in natural language processing while retaining complete intellectual responsibility for the scientific and technical contributions of this work.

```
Your task is to convert each Object mentioned in a given sentence into a corresponding instruction, and all the resulting instructions are
output as "Describe more details about the [Object]". Ensure your instructions do not cover the raw question, options, or thought
process of answering the instructions. You should ignore the Objects that appear in some inferences, such as the sentences that begins
with 'it might be' or 'there are probably'.
Sentence:
The image depicts a man in a suit and tie jumping in the air above a bed in a bedroom
Instructions:
Describe more details about the man.
Describe more details about the suit.
Describe more details about the tie.
Describe more details about the bed.
Describe more details about the bedroom.

Sentence:
The train appears to be the main subject of the image, showcasing its sleek design and modern appearance
Instructions:
Describe more details about the train.

Sentence:
The table has a few other items on it, including a camera, a jar of jam, and a spoon, suggesting that there might be some people ready to
eat
Instructions:
Describe more details about the table.
Describe more details about the camera.
Describe more details about the jam.
Describe more details about the spoon.

Sentence:
The text "You see the world as you are!" is a playful and thought-provoking statement, encouraging viewers to appreciate their unique
qualities and perspectives
Instructions:
Describe more details about the text.

Sentence:
1. **Preheat the Oven**: Preheat your oven to 350\u00b0F (175\u00b0C).
Instructions:
Describe more details about the oven.
Describe more details about the preheat temperature.

Sentence:
{}
Instructions:
```

Figure 9: Prompt $T_{ict}$ used in ScaleCap to generate object instructions.

## F DATASET PROCESSING

**Data Source and Processing.** In collecting images for our dataset, we primarily focus on two aspects: diversity and richness of image content. Given that the ShareGPT4V-100k already includes a wide range of categories, such as artworks, landmarks, etc., it inherently offers a certain level of diversity. Therefore, we opt to directly incorporate these images into our dataset. To further enhance the dataset's diversity and to obtain more content-rich images, we additionally select 350k images from the LAION-5BLAION (2022) dataset. The LAION-5B dataset is sourced from publicly available content on the internet, encompassing a wide range of subjects, styles, and domains due to its web-based origin. This ensures a high level of diversity in the collected data. During filtering, we retain only images with high resolution and moderate complexity. During the image selection process, we filter out images with a short-edge resolution of less than 600 pixels to preserve the richness of visual content. To further ensure the complexity and informativeness of the images, we employed (Feng et al., 2023) to score image complexity. Images were filtered based on a complexity range of [0.4, 0.8], which helped exclude both overly simplistic and excessively complex images, thereby maintaining the overall quality of the dataset. During the image selection process, we also conduct manual sampling to filter out potentially harmful images.

**Caption Model Selection.** As previously mentioned, ScaleCap leverages the collaboration between a Vision Language Model and a Large Language Model to generate high-quality captions. For LVLM, as we discussed above, a small model is capable of capturing visual content, so we use Qwen2-VL-7B by default. When it comes to the LLM, the question-raising task is relatively simple, while the integration of complex and abundant visual information within thousands of tokens requires advanced reasoning capability, so we resort to Qwen2-72B based on an empirical study.

# G  PRETRAINING DETAILS

**Training Setting.** We follow the training strategy consistent with ShareGPT4V, dividing the overall training process into three stages. (1) Initial Pretraining Stage. In this stage, we train the projector from scratch using the BLIP-558K dataset for pre-alignment. The objective is to establish an initial mapping between visual and textual modalities. We adopt a learning rate of 1e-3 and a batch size of 256. (2) Further Pretraining Stage. We initialize the projector with the weights obtained from the initial pretraining stage. Both the projector and the LLM are jointly optimized using high-quality image caption in this stage. This stage leverages high-quality image-text pairs to achieve effective alignment of the visual features extracted by the vision encoder and the corresponding text features in the LLM. We set the learning rate to 4e-5 and the batch size to 256. (3) Instruction-tuning Stage. During this stage, we use open-source datasets Open-LLaVA-NeXT-Instruct-1M(Chen & Xing, 2024) to finetune the projector and the LLM. Previous research has demonstrated the effectiveness of using open-source datasets such as Open-LLaVA-NeXT-Instruct-1M, which includes diverse data sources such as DocVQA and SynDog-EN. We adopt this dataset directly for instruction tuning. During this stage, both the projector and the LLM are jointly fine-tuned. A learning rate of 2e-5 and a batch size of 128 are used.

**Baselines.** (1) Vanilla. We select the model trained through only two stages, namely the Initial Pretraining Stage and the Instruction-tuning Stage, as our basic baseline. This model does not undergo any additional further pretraining and corresponds to the default configuration of the original LLaVA-NeXT. (2) ShareGPT4V-450k. For fair comparison, the amount of data is strictly aligned with that of ScaleCap-450k. Specifically, ShareGPT4V-450k consists of ShareGPT4V-100k and 350k samples randomly selected from ShareGPT4V-PT. (3) DenseFusion-450k. It is a subset randomly sampled from DenseFusion-1M, which is a high-quality dataset constructed using diverse perception experts as image priors to provide explicit information on visual elements. During the training, we use different datasets only in the Further Pretraining Stage, while keeping other settings exactly the same to ensure a fair comparison.

Our experiments are conducted on 8 A100 GPUs. The initial pretraining stage using BLIP-558K took approximately 2.5 hours, the further pretraining using ScaleCap-450K required 13.5 hours, and the final instruction-tuning stage took 17.2 hours.

# H  CURRENT LANDSCAPE OF MULTIMODAL MODELS

Multimodal learning has accelerated rapidly in the past two years, driven by systems that couple language with perception and action at scale (Liu et al., 2023; Bai et al., 2025; Li et al., 2025; 2026; Xing et al., 2025; 2024; Liu et al., 2025; Carion et al., 2025; Qian et al., 2025; 2024b; Ding et al., 2023; Qian et al., 2023). These advances(Zhang et al., 2023; 2025; Qi et al., 2025a;b; Qian et al., 2024a) have produced broad empirical breakthroughs across benchmarks and real-world tasks. Compared to earlier eras that focused mainly on images paired with text, today's models are larger, trained on more diverse corpora, and natively support additional modalities such as video and audio.

# I  USER STUDY INSTRUCTIONS

To directly compare the richness and accuracy of captions generated by different LVLMs, we use the captions produced by each model as prompts for FLUX to generate corresponding images. In the following questionnaire, please rank the images based on their similarity to the original image.

