# OpenReview forum: "ScaleCap: Scalable Image Captioning via Dual-Modality Debiasing"
_ICLR.cc/2026/Conference — ICLR 2026 Poster_

### Official Review · Reviewer_VAQo · 2025-10-21

**Soundness:** 3
**Presentation:** 3
**Contribution:** 2
**Rating:** 4
**Confidence:** 5

**Summary:**

This paper proposes ScaleCap, a detailed image captioning pipeline that uses multiple models, including an LLM and an LVLM. It first generates an initial caption and identifies golden sentences, highly likely non-hallucinatory sentences within the generated caption, based on contrastive sentence ratings. Contrastive sentence rating is a strategy for evaluating the factual precision of a sentence by comparing output token probabilities between multimodal decoding and language-only decoding. Then, ScaleCap further obtains visual information by iteratively requesting detailed descriptions for each object mentioned in the golden sentences (referred to as heuristic question-answering module). By alternating between contrastive sentence rating and heuristic question answering, ScaleCap collects reliable sentences and finally summarizes them using an LLM. The authors demonstrate the effectiveness of ScaleCap through reconstruction and pretraining experiments.

**Strengths:**

1. This paper is clear and well-organized overall.
2. The problem addressed in the paper is interesting and timely.

**Weaknesses:**

1. The authors use image reconstruction and pretraining experiments to demonstrate the effectiveness of the proposed method. However, there are still many evaluation approaches remaining for assessing detailed image captions. For example, GPT-based methods [1,2] and QA-based methods [3] could be leveraged. I strongly encourage the authors to incorporate these additional evaluation metrics.
2. Several previous studies have proposed detailed image captioning systems that involve multiple deep learning models [3, 4]. Comparisons and discussions with these works are needed to better position the proposed method within the existing literature.

[1] Petryck et al., "ALOHa: A New Measure for Hallucination in Captioning Models"
[2] Chan et al., "CLAIR: Evaluating Image Captions with Large Language Models"
[3] Lee et al., "Toward Robust Hyper-Detailed Image Captioning: A Multiagent Approach and Dual Evaluation Metrics for Factuality and Coverage"
[4] Get et al., "Visual Fact Checker: Enabling High-Fidelity Detailed Caption Generation"

**Questions:**

oes the proposed method outperform existing methods across multiple evaluation metrics?

---

> ### Author Response · Authors · 2025-11-21
> **Reponse to Reviewer VAQo**
>
> We greatly appreciate the reviewer’s insightful comments, careful evaluation, and positive feedback. In response, we have carried out additional experiments and revised the manuscript accordingly. Our point-by-point replies are provided below.
>
> > **W1: The authors use image reconstruction and pretraining experiments to demonstrate the effectiveness of the proposed method. However, there are still many evaluation approaches remaining for assessing detailed image captions. For example, GPT-based methods [1,2] and QA-based methods [3] could be leveraged. I strongly encourage the authors to incorporate these additional evaluation metrics.**
>
> Thank you for this very helpful suggestion! As you point out, our original evaluations focused primarily on downstream uses of the captions (reconstruction and pretraining). Following your recommendation, we have added GPT-based and QA-style caption quality evaluations to more directly assess detailed caption quality.
>
> Concretely, we evaluate ScaleCap using ALOHa, CLAIR, and the Factuality/Coverage metrics from CapMAS. ALOHa relies on ground-truth (GT) captions as references, but the original GT captions of the HAT dataset are short, low-quality descriptions produced by older models such as BLIP (e.g., "A picture of goats eating grass"), which makes the evaluation somewhat outdated for our setting. To obtain a fairer and more informative reference, we regenerate GT captions using Qwen3VL-235B, one of the strongest current open-source LVLMs, while keeping the ALOHa scoring protocol unchanged. For all benchmarks, we strictly follow their official evaluation procedures and use a strong GPT-based evaluator (GPT5-mini) where required.
>
> | Caption Model | ALOHa | CLAIR | Factuality | Coverage | avg |
> | :--- | :--- | :--- | :--- | :--- | :--- |
> | Qwen2-VL-7B | 64.1 | 75.3 | 76.6 | 63.8 | 70.0 |
> | Qwen2-VL-72B | 65.8 | 74.9 | 81.8 | 66.2 | 72.2 |
> | ScalCap | 66.9 | 76.8 | 79.4 | 73.1 | 74.1 |
>
> Across these metrics, ScaleCap consistently outperforms both Qwen2-VL-7B and Qwen2-VL-72B: on average, ScaleCap improves over Qwen2-VL-7B by 4.1% and over Qwen2-VL-72B by 1.9%. In particular, on the Coverage metric proposed in CapMAS, ScaleCap exceeds Qwen2-VL-7B and Qwen2-VL-72B by 9.3% and 6.9%, respectively, indicating that our Heuristic Question Answering module effectively uncovers more image details, while the Contrastive Sentence Rating module maintains factual quality. We will incorporate these new evaluations and the corresponding results table into the revised manuscript to address this point.
>
> > **W2 & Q1: Several previous studies have proposed detailed image captioning systems that involve multiple deep learning models [3, 4]. Comparisons and discussions with these works are needed to better position the proposed method within the existing literature.**
>
> We thank the reviewer for this valuable suggestion. Since the official implementations of these systems are not publicly available, we approximate CapMAS and VFC as faithfully as possible based on their papers and released prompts, and then compare them to ScaleCap under a unified setting. Concretely, we use Qwen2VL-7B as the captioner and Qwen2-72B as the LLM in both CapMAS and VFC, and adopt Grounding DINO (box_threshold = 0.35) as the open-vocabulary detector.
>
> | Caption Model | ALOHa | CLAIR | Factuality | Coverage | avg |
> | :--- | :--- | :--- | :--- | :--- | :--- |
> | Qwen2-VL-7B | 64.1 | 75.3 | 76.6 | 63.8 | 70.0 |
> | VFC | 65.5 | 76.1 | 78.5 | 63.1 | 70.8 |
> | CapMAS | 67.3 | 76.4 | 78.9 | 63.3 | 71.5 |
> | Qwen2-VL-72B | 65.8 | 74.9 | 81.8 | 66.2 | 72.2 |
> | ScaleCap | 66.9 | 76.9 | 79.4 | 73.1 | 74.1 |
>
> The results (Table above) show that both CapMAS and VFC improve over the base Qwen2-VL-7B captioner, but still fall noticeably short of ScaleCap. In particular, while their internal verifier modules can increase the Factuality metric, they tend to reduce Coverage, likely because verification errors suppress valid objects and neither method explicitly enriches visual detail beyond the initial caption. In contrast, ScaleCap’s heuristic question answering explicitly discovers additional object- and attribute-level details, and its contrastive sentence rating filters hallucinations without sacrificing coverage, leading to stronger overall caption quality.
>
> We have (1) added these quantitative comparisons to the experimental section, and (2) expanded the related work discussion to better position ScaleCap relative to CapMAS and VFC, emphasizing the differences in design philosophy (model verifier pipelines vs. our dual-modality debiasing strategy designed for scalable, detail-rich caption generation).
>
> Thanks again for your constructive comments and meticulous review, which have helped us significantly improve the paper's quality! Please don't hesitate to let us know if there are any additional clarifications or experiments that we can offer.

---

> > ### Comment · Reviewer_VAQo · 2025-11-26
> >
> > I appreciate the authors’ hard work in addressing my concerns. I would appreciate it even more if the authors could answer my follow-up question: the proposed method appears to be a well-designed combination of prior works. Could the authors summarize the novel contributions and newly discovered findings in this work?

---

> > > ### Author Response · Authors · 2025-11-26
> > > **Reponse to Reviewer VAQo**
> > >
> > > We sincerely thank the reviewer for acknowledging the effort put into our previous response and for the opportunity to further clarify the novelty of our work. Our primary novelty and significant contribution lie in identifying (1) two major flaws in existing captioning methodologies that prior works have overlooked, and exactly based on these insights, (2) designing new, specialized captioning paradigms that yield substantial improvements. To the best of our knowledge, these observations and our methodological designs have not been presented in any previous work. Below, we detail our newly discovered findings and our specialized designs.
> > >
> > > (1) We discovered that existing online contrastive decoding methods (e.g., VCD) suffer from a severe 'fluency dilemma.' For instance, an original output such as:
> > >
> > > > *A man is sitting on a park bench reading a book.*
> > >
> > > After applying an online contrastive decoding method, often becomes:
> > >
> > > > *There is a person in the park. The person is reading.*
> > >
> > > Such methods aggressively interfere with the model's output in an online manner, resulting in disfluent and unnatural sentences. Our key observation reveals that such online, token-level contrastive decoding methods disrupt the inherent language distribution of the LLM, often leading to broken syntax and incoherent outputs. Precisely because we identified this critical issue in existing contrastive decoding strategies, we fundamentally altered this paradigm by shifting from online token-level interference to offline sentence-level verification. After the model generates a complete response, we apply a carefully designed scoring strategy to filter hallucinations at the sentence level. Our method not only generates fluent outputs but also significantly improves factuality (CHAIRs 33.6 vs. 46.8 for VCD).
> > >
> > > (2) Additionally, considering the visual infomation coverage in LVLM caption, we identified a phenomenon that has not been previously discussed: LVLMs consistently fail to comprehensively express certain visual details. Our experiments further revealed a counter-intuitive insight: LVLMs do not lack perceptual capability; rather, they fail due to suboptimal instruction design. To address this performance issue in LVLMs, we propose an innovative iterative question–answering strategy to progressively elicit more details. We leverage an LLM to construct simple but effective instructions to stimulate the LVLM's capabilities, excavating missed visual details, and finally utilizing the LLM's powerful summarization capabilities to integrate all visual information.
> > >
> > > Thank you again for your valuable feedback and for your responsible attitude toward the entire review community! We are happy to provide further clarification or discuss any remaining questions you may have.

---

### Official Review · Reviewer_jUSR · 2025-10-26

**Soundness:** 3
**Presentation:** 3
**Contribution:** 2
**Rating:** 4
**Confidence:** 4

**Summary:**

This paper introduces ScaleCap, a pipeline designed to generate comprehensive and detailed image captions for pretraining VLM. The authors identify multimodal bias and linguistic bias as key challenges. ScaleCap addresses these through Heuristic Question Answering and Contrastive Sentence Rating. Using ScaleCap, the authors create the ScaleCap-450K dataset  and show that pretraining LVLMs on this dataset consistently improves performance compared to using datasets like ShareGPT4V and DenseFusion.

**Strengths:**

1.The paper is generally well-written and clearly structured, making the proposed pipeline and experiments easy to follow.

2.The work proposes a complete pipeline aimed at improving the quality, detail, and factuality of LVLM-generated captions.

3.Experiments effectively demonstrate that the ScaleCap-450K dataset, generated by the proposed pipeline, leads to superior LVLM pretraining outcomes compared to existing large-scale caption datasets like ShareGPT4V-450k and DenseFusion-450k.

**Weaknesses:**

1.The core components of ScaleCap, namely Heuristic Question Answering and Contrastive Sentence Rating, appear to have limited novelty, primarily combining or refining existing techniques rather than introducing fundamentally new concepts.

2.The proposed ScaleCap pipeline suggests a high computational cost for generating each caption, involving multiple model inference stages including initial captioning, filtering, question generation, iterative question answering, answer filtering, and final integration using a large LLM. This multi-step process appears resource-intensive, potentially limiting its practical scalability.

**Questions:**

1.Could the authors quantify the computational cost—such as average time per caption or total GPU hours required to generate a single caption using the full ScaleCap pipeline, and how does this compare to baseline caption generation costs?

2.While the paper compares pretraining benefits against other datasets, how does the quality of ScaleCap captions or the performance of models pretrained on them compare against models directly improved using training-based hallucination mitigation techniques like RLAIF-V or LLaVA-RLHF?

3.Considering the introduction's potentially restrictive view on tool-based captioning , could the ScaleCap pipeline itself benefit from integrating specific tools, perhaps for grounding question generation or verifying answers, rather than solely relying on the VLM and CSR

---

> ### Author Response · Authors · 2025-11-21
> **Reponse to Reviewer jUSR (1/3)**
>
> We sincerely appreciate the reviewer’s thoughtful comments, thorough evaluation, and positive feedback. In response, we have carried out additional experiments and revised the manuscript accordingly. Detailed replies are provided below.
>
> > **W1: The core components of ScaleCap, namely Heuristic Question Answering and Contrastive Sentence Rating, appear to have limited novelty, primarily combining or refining existing techniques rather than introducing fundamentally new concepts.**
>
> We thank the reviewer for this comment and the opportunity to clarify our contributions. While ScaleCap indeed builds on existing building blocks (contrastive decoding, LVLM Q&A), our core novelty lies in (i) identifying two structural flaws in how these techniques are currently used, and (ii) proposing new formulations that directly address them and lead to substantial empirical gains.
>
> 1) The Contrastive Sentence Rating module is explicitly designed to resolve what we term the "fluency dilemma" in online contrastive decoding (e.g., VCD). Existing methods intervene at the token level during generation, which we show systematically distorts the language model’s native distribution and often produces disfluent, generic sentences, even when hallucinations are reduced. Our approach fundamentally changes the paradigm: instead of online token-level interference, we generate fluent candidate sentences with the base model and then perform offline, sentence-level contrastive verification to prune hallucinations. This design both preserves fluency and substantially improves factuality (e.g., CHAIRs 33.6 vs. 46.8 for VCD), which, to our knowledge, has not been explored in prior contrastive decoding work.
>
> 2) Through Heuristic Question Answering, we identify and leverage a previously under-explored property of LVLMs: their failure to mention many visual details is often due to instruction design rather than perceptual blindness. Our analysis shows that LVLMs can recognize these details when queried appropriately, but standard caption prompts do not elicit them. Based on this insight, we introduce an iterative, budget-controllable question-answering strategy that systematically surfaces missing object- and relation-level information and injects it into the caption. This targeted, instruction-driven enrichment of visual detail coupled with dual-modality debiasing, to the best of our knowledge, is not present in earlier multi-model captioning pipelines.
>
> > **W2: The proposed ScaleCap pipeline suggests a high computational cost for generating each caption, involving multiple model inference stages including initial captioning, filtering, question generation, iterative question answering, answer filtering, and final integration using a large LLM. This multi-step process appears resource-intensive, potentially limiting its practical scalability.**
>
> We appreciate the reviewer’s concern about the computational cost of our pipeline and agree that efficiency is an important consideration. We address this concern in the following three ways:
>
> 1) ScaleCap is explicitly designed with a **tunable performance–budget trade-off**. As shown in our caption scaling analysis (Figure 7a), the number of heuristic instructions directly controls the level of detail: users can select a smaller instruction budget to obtain reasonably detailed captions at substantially lower cost, or increase the budget only when maximal richness is required.
>
> 2) Our primary goal is to construct high-quality caption datasets for LVLM pre-training in an **offline setting**, rather than to support real-time captioning. In this context, the multi-stage pipeline represents a one-time investment: once generated, the ScaleCap captions can be reused indefinitely for training many different models. As shown in Table 1, pre-training on ScaleCap-450K yields performance gains that are considerably larger than those obtained from prior datasets, suggesting that the resulting benefits justify the additional generation cost. **We also bear the full cost of building ScaleCap-450K** and **will release the dataset** to the community, so the research community can directly benefit without incurring this expense.
>
> 3) The components of ScaleCap are **highly parallelizable**. In practice, each stage (initial captioning, golden sentence extraction, question generation, iterative Q&A, filtering, and integration) can be batched and distributed across GPUs. For example, by first processing all images for initial captions, then all for golden sentences, and so on, while using inference accelerators such as vLLM. This pipeline structure allows us to maximize hardware utilization and amortize latency, making ScaleCap practical for large-scale offline dataset construction.

---

> ### Author Response · Authors · 2025-11-21
> **Reponse to Reviewer jUSR (2/3)**
>
> > **Q1: Could the authors quantify the computational cost—such as average time per caption or total GPU hours required to generate a single caption using the full ScaleCap pipeline, and how does this compare to baseline caption generation costs?**
>
> We thank the reviewer for raising this important concern regarding computational overhead. For fair comparison, we standardized using vLLM to perform inference and generate captions for 1,000 images on a single node (8 A100 GPUs). All the results are shown in the figure below.
>
> | Caption Method | time | Avg Prism Score |
> | :--- | :--- | :--- |
> | Qwen2VL-7B | 3.3min | 54.1 |
> | Qwen2VL-72B | 50.7min | 56.0 |
> | ScaleCap | 256min | 58.2 |
>
> We also include the Prism Score to simultaneously compare caption quality and inference cost. We can observe that while Qwen2VL-72B is over 10 times slower than Qwen2VL-7B, it only yields a 1.9% gain in Prism Score. In contrast, ScaleCap achieves a 2.2% improvement over Qwen2VL-72B, with only five times the inference time. When viewed through the lens of the classic budget-performance scaling curve, ScaleCap’s performance far exceeds expectations, effectively avoiding the plateau of diminishing marginal utility.
>
> > **Q2: While the paper compares pretraining benefits against other datasets, how does the quality of ScaleCap captions or the performance of models pretrained on them compare against models directly improved using training-based hallucination mitigation techniques like RLAIF-V or LLaVA-RLHF?**
>
> We thank the reviewer for raising this insightful question regarding training-based hallucination mitigation techniques. To provide a comprehensive comparison against training-based hallucination mitigation techniques, we evaluate our method not only against RLAIF-V and LLaVA-RLHF, but also against other DPO-series algorithms, specifically HADPO and mDPO, and the contrastive tuning method HALVA. The results are presented below.
>
> |  | Training data size | CHAIRS↓ | CHAIR I↓ |
> | :--- | :--- | :--- | :--- |
> | LLaVA-v1.5 7B | - | 48.8 | 13.9 |
> | +LLaVA-RLHF | 122k | 47.1 | 13.3 |
> | +HALVA | 21.5k | 41.4 | 11.7 |
> | +HADPO | 6k | 37.9 | 11.9 |
> | +mDPO | 10k | 35.7 | 9.8 |
> | +RLAIF-V | 16k | 16.0 | 3.7 |
> | +Ours | Training-free | 33.6 | 11.3 |
>
> The findings show that our method, even without any dedicated training, surpasses most data-intensive training approaches in reducing hallucinations, second only to RLAIF-V. This strongly demonstrates the high effectiveness of our proposed Contrastive Sentence Rating module.

---

> ### Author Response · Authors · 2025-11-21
> **Reponse to Reviewer jUSR (3/3)**
>
> > **Q3: Considering the introduction's potentially restrictive view on tool-based captioning , could the ScaleCap pipeline itself benefit from integrating specific tools, perhaps for grounding question generation or verifying answers, rather than solely relying on the VLM and CSR**
>
> We appreciate this insightful suggestion and agree that tool integration is a promising direction for extending ScaleCap. To directly probe this, we experimented with adding a Grounding DINO module after the Contrastive Sentence Rating stage: for each LLM-generated instruction, Grounding DINO checks whether the mentioned object is present in the image, and instructions whose objects are not grounded are discarded. On the Prism evaluation setting, Qwen2VL-7B, Qwen2VL-72B, and our original ScaleCap achieve average scores of 56.2, 58.2, and 60.3, respectively, while "ScaleCap + Grounding DINO" reaches 59.4 (MMVet: 58.7, MMStar: 50.3, ChartQA: 69.1). The gains on natural-image benchmarks are marginal and ChartQA performance noticeably drops, likely because (i) our CSR already removes most hallucinations, and (ii) Qwen2VL provides strong built-in grounding comparable to current generic detectors, whereas Grounding DINO is not well-suited to chart-style inputs.
>
> | Caption Method | MMVet | MMStar | Chart QA | Avg |
> | :--- | :--- | :--- | :--- | :--- |
> | Qwen2VL-7B | 53.5 | 46.6 | 68.6 | 56.2 |
> | Qwen2VL-72B | 56.9 | 47.5 | 70.2 | 58.2 |
> | ScaleCap | 58.8 | 49.5 | 72.5 | 60.3 |
> | ScaleCap with GroudingDINO | 58.7 | 50.3 | 69.1 | 59.4 |
>
> These results suggest that, in our current setting, general-purpose grounding tools bring limited incremental benefit over a carefully designed dual-modality debiasing pipeline. That said, we view ScaleCap as a flexible framework rather than as opposed to tools: more specialized tools (e.g., domain-specific detectors, OCR, or even web search for up-to-date knowledge) can be incorporated to further enhance grounding or enrich specific types of details (e.g., identifying a toy as a particular designer brand, 'This plush toy is the recently popular designer toy, LABUBU.').
>
> We have clarified this nuance in the discussion in the appendix: our critique is of tool-centric pipelines that rely heavily on brittle, domain-specific tools, not of tools per se, and we see tool integration as a natural future extension of ScaleCap when suitable tools are available.
>
> Thanks again for your constructive comments and meticulous review, which have helped us significantly improve the paper's quality! Please don't hesitate to let us know if there are any additional clarifications or experiments that we can offer.

---

> ### Author Response · Authors · 2025-11-27
> **Official Comment by Authors**
>
> Dear Reviewer jUSR,
>
> We sincerely appreciate your time and effort in providing such meticulous reviews and insightful comments.
>
> Could you please kindly let us know if our rebuttal has addressed your concerns?
>
> If you have any further questions regarding our work, we would be delighted to address them.
>
> Best regards,
>
> Authors

---

### Official Review · Reviewer_x2KT · 2025-10-29

**Soundness:** 2
**Presentation:** 3
**Contribution:** 3
**Rating:** 6
**Confidence:** 3

**Summary:**

The paper introduces *ScaleCap, a scalable image-text paired data distillation pipeline designed to produce long, instance-complete, and visually grounded captions from LVLM without external detectors/tools. ScaleCap has two core components:
(1) Heuristic Question Answering (HQA): an LLM generates targeted, object/attribute/position questions from an initial caption; a LVLM answers them to surface missing details under a controllable budget.
(2) Contrastive Sentence Rating (CSR): an offline filter that compares token probabilities with vs. without the image and retains sentences whose “critical tokens” are better supported by the image than by language priors, aiming to suppress hallucinations.
Using this pipeline, the authors build ScaleCap-450K, a 450k-image caption dataset sourced mainly from LAION and ShareGPT4V images with resolution/complexity filtering.

**Strengths:**

1. The two identified blockers map neatly to HQA (add missing visual facts) and CSR (filter unsupported text). The mechanism is easy to reason about and implement with standard LVLM/LLM primitives.
2. The observation that 7B-class LVLMs are often sufficient for perception while larger LLMs help during long-context integration provides concrete guidance for cost-constrained systems.
3. Improvements appear across multiple LVLM backbones/settings, suggesting the dataset’s benefits aren’t model-specific.

**Weaknesses:**

1. The core claim that balanced, instance-complete detail drives the gains isn’t cleanly isolated from caption length or sheer verbosity. Controlled studies (equal length across methods; fixed token budgets redistributed among object/attribute/position details) are missing.
2. CSR accepts sentences based on a max-over-critical-tokens Δ probability threshold. This may be unstable (single-token spikes; POS-tagging noise). Robustness analyses (pooling variants, τ-sweeps, cross-dataset calibration) are not provided.

**Questions:**

1. For frontier models, does the way ScaleCap introduces more prior information help the LVLM itself in the pipeline, such as fine-tuning the LVLM with generated data?
2. If CSR is computed with a different LVLM than the one used to answer HQA (or than the pretraining backbone), do conclusions hold?
3. A clear pseudocode algorithm will be more helpful for reading and understanding.

---

> ### Author Response · Authors · 2025-11-21
> **Reponse to Reviewer x2KT (1/3)**
>
> We are truly grateful to the reviewer for the valuable comments, thorough evaluation, and encouraging remarks. In response, we have carried out additional experiments and revised the manuscript accordingly. Our detailed replies are presented below.
>
> > **W1: The core claim that balanced, instance-complete detail drives the gains isn’t cleanly isolated from caption length or sheer verbosity. Controlled studies (equal length across methods; fixed token budgets redistributed among object/attribute/position details) are missing.**
>
> We sincerely appreciate the reviewer for raising this critical point regarding the distinction between useful detail and mere verbosity. However, adding more visual details in the caption will inevitably increase its length, and forcefully truncating all captions to the same length for comparison would not be very reasonable. Therefore, it is not easy to compare different methods under a strictly fixed token budget.
>
> **(1) Verbosity comparison between captions**
>
> Due to the difficulty of strictly fixing the token budget,  we employed a more direct evaluation method, utilizing GPT-5 to assess Verbosity.
> We randomly sampled 200 images from MMStar, ChartQA, InfoVQA, and MMMU, and generated captions using Qwen2-VL-7B, Qwen2-VL-72B, and our method, ScaleCap. We then prompted GPT-5 to evaluate caption quality on a scale of 1 to 5 based on four distinct criteria: **Succinctness Degree (measuring verbosity), Detail Coverage, Content Accuracy, and Expression Fluency**.
>
> | Caption Model | Succinctness Degree | Detail Coverage | Content Accuracy | Expression Fluency | Avg |
> | :--- | :--- | :--- | :--- | :--- | :--- |
> | Qwen2-VL-7B | 4.05 | 3.51 | 3.55 | 4.20 | 3.83 |
> | Qwen2-VL-72B | 4.13 | 3.91 | 4.04 | 4.34 | 4.11 |
> | ScaleCap | 4.07 | 4.45 | 4.19 | 4.51 | 4.31 |
>
> The results demonstrate that ScaleCap achieves Succinctness scores comparable to the Qwen2-VL 7B and 72B. This suggests that the effective information delivered per token is similar across models. Crucially, however, ScaleCap significantly outperforms the baselines in Detail Coverage and Content Accuracy. This indicates that while maintaining a similar level of succinctness (avoiding verbosity), ScaleCap successfully encapsulates **significantly more accurate and comprehensive visual information**.
>
> **(2) Budgets redistributed among object/position details**
>
> Although directly limiting the token length is difficult, we can effectively control **the budget for object/position details by adjusting the number of Object Instructions and Position Instructions**.
>  Based on this idea, we conduct controlled studies on how the budget distribution affects caption quality. We randomly sample a fixed number of instructions from all Object Instructions, and perform the corresponding operation for Position Instructions. The experimental results are as follows.
>
> | Number of Object instruction | Number of Postion instruction | MMVet | MMStar | ChartQA | Avg |
> | :--- | :--- | :--- | :--- | :--- | :--- |
> | 20 | 0 | 53.9 | 47.9 | 68.7 | 56.8 |
> | 10 | 10 | 55 | 48.3 | 69.2 | 57.5 |
> | 0 | 20 | 54.1 | 46.5 | 64.4 | 55.0 |
> | Unlimited | Unlimited | 58.8 | 49.5 | 72.5 | 60.3 |
>
> The results indicate that relying solely on either Object instructions or Position instructions leads to a significant performance drop. The best results are achieved when the two are balanced, suggesting that Object instructions and Position instructions work closely together to uncover more image detail.

---

> ### Author Response · Authors · 2025-11-21
> **Reponse to Reviewer x2KT (2/3)**
>
> > **W2: CSR accepts sentences based on a max-over-critical-tokens Δ probability threshold. This may be unstable (single-token spikes; POS-tagging noise). Robustness analyses (pooling variants, τ-sweeps, cross-dataset calibration) are not provided.**
>
> We sincerely thank the reviewer for their insightful feedback regarding the robustness of the CSR mechanism. To address this concern, we have conducted additional experiments to examine the sensitivity of the threshold parameter ($\tau$) and cross-dataset performance.
>
> **(1) Sensitivity Analysis of Threshold $\tau$**
>
> To evaluate the stability of our hyperparameter, we collected a subset of 200 images from the MMStar dataset. We utilized Qwen2VL-7B to generate the initial captions, resulting in a total of 2,756 sentences. CSR was then employed to detect hallucinations in each sentence. To establish the ground truth, we used GPT-5 (with access to the original images) to verify whether each sentence factually contained hallucinations. The calculated results are presented in the table below:
>
> | threshold | -1 | 0 | 1 |
> | :--- | :--- | :--- | :--- |
> | Recall | 73.8% | 73.5% | 72.8% |
> | Precision | 64.2% | 65.5% | 65.8% |
>
> These results demonstrate that our hyperparameter, threshold $\tau$, exhibits strong robustness. When varying the value across $\{-1, 0, 1\}$, the overall performance remains relatively stable. The general trend shows that when $\tau = -1$, we are performing a more stringent selection, leading to an increase in Recall and a decrease in Precision; conversely, when $\tau = 1$, the selection is more lenient, resulting in a decrease in Recall and an increase in Precision. We selected $\tau = 0$ for our main experiments to achieve an optimal balance between Recall and Precision.
>
> **(2) Cross-dataset Performance**
>
> To further assess robustness across different domains, we collected an additional 200 images from SeedBench. Following the same process established above, we used Qwen2VL-7B for caption generation and GPT-5 for ground-truth verification. We applied CSR with the fixed threshold $\tau = 0$.
>
> | Image source | MMStar | SeedBench |
> | :--- | :--- | :--- |
> | Recall | 73.5% | 72.6% |
> | Precision | 65.5% | 64.9% |
>
> The results indicate that performance remains consistent when transferring from MMStar to SeedBench. This confirms that our CSR algorithm maintains stability and effectiveness even when applied to datasets with significant domain differences.
>
> > **Q1: For frontier models, does the way ScaleCap introduces more prior information help the LVLM itself in the pipeline, such as fine-tuning the LVLM with generated data?**
>
> We thank the reviewer for this thoughtful suggestion. Intuitively, we agree that frontier LVLMs could benefit from additional prior information introduced by ScaleCap-style data, especially when used at the SFT stage.
>
> However, in our setting we face two practical limitations. First, we do not have access to the full Qwen2-VL pre-training/SFT corpus or the exact fine-tuning recipe (e.g., data mixture, learning rate schedule, training epoches), which makes it difficult to cleanly integrate ScaleCap-450K into the official SFT pipeline and fairly assess the effect. Second, we attempted a more conservative proxy experiment by converting ScaleCap captions into QA-style pairs and performing an additional SFT stage on Qwen2-VL with a learning rate of 5e-6. However, this native fine-tuning leads to a noticeable drop in performance on the original evaluation tasks, suggesting that the model overfits to the caption-style supervision and partially forgets its broader skills.
>
> These observations indicate that simply “stacking” ScaleCap data at the end of SFT is not sufficient, and that carefully designed multi-task or multi-stage training strategies (with access to the full training mixture and hyperparameters) are likely needed to unlock the full benefit. We will highlight integrating ScaleCap data into frontier LVLM training as an important direction for future work.

---

> ### Author Response · Authors · 2025-11-21
> **Reponse to Reviewer x2KT (3/3)**
>
> > **Q2: If CSR is computed with a different LVLM than the one used to answer HQA (or than the pretraining backbone), do conclusions hold?**
>
> We thank the reviewer for this constructive suggestion. Using an LVLM that is completely different from CSR for HQA is entirely feasible. In fact, mixing different LVLMs could theoretically capture a broader range of visual information. Therefore, we replace the LVLM used for HQA after CSR with InternVL3-8B, other models remain unchanged. The experimental results are shown below.
>
> | Caption Method | MMVet | MMStar | Chart QA | Avg |
> | :--- | :--- | :--- | :--- | :--- |
> | ScaleCap | 58.8 | 49.5 | 72.5 | 60.3 |
> | ScaleCap-InternVL3-8B | 59.5 | 50.1 | 74.1 | 61.2 |
>
> There is a noticeable overall improvement, which stems from both the stronger performance of InternVL3-8B itself and the increased diversity of visual information provided by this approach. InternVL3-8B can supplement Qwen2-VL with additional information that it would otherwise overlook.
>
> > **Q3: A clear pseudocode algorithm will be more helpful for reading and understanding.**
>
> We thank the reviewer for this valuable suggestion. Following your recommendation, we have included a structured pseudocode algorithm in the Appendix to facilitate reading and understanding.
>
> Thanks again for your constructive comments and meticulous review, which have helped us significantly improve the paper's quality! Please don't hesitate to let us know if there are any additional clarifications or experiments that we can offer.

---

> ### Author Response · Authors · 2025-11-27
> **Official Comment by Authors**
>
> Dear Reviewer x2KT,
>
> We sincerely appreciate your time and effort in providing such meticulous reviews and insightful comments.
>
> Could you please kindly let us know if our rebuttal has addressed your concerns?
>
> If you have any further questions regarding our work, we would be delighted to address them.
>
> Best regards,
>
> Authors

---

### Official Review · Reviewer_wjkS · 2025-11-04

**Soundness:** 3
**Presentation:** 3
**Contribution:** 2
**Rating:** 6
**Confidence:** 3

**Summary:**

The paper proposes ScaleCap, a captioning pipeline that iteratively enriches captions while reducing hallucinations by combining (i) heuristic question answering and (ii) contrastive sentence rating. A tunable scale budget controls how many questions are asked, trading cost for detail. Using ScaleCap, the authors build ScaleCap-450K and show consistent gains across 11 benchmarks, plus benefits in Prism-style perception tests and an image-reconstruction study. The author argues that the approach improves informativeness and alignment even with smaller LVLMs.

**Strengths:**

- Clear, modular method that targets two real failure modes. The offline contrastive, sentence-level rating is a neat way to down-weight language priors without destabilizing decoding.
- Strong empirical coverage and breadth. Pretraining with ScaleCap-450K beats ShareGPT4V-450K and DenseFusion-450K on most of the 11 benchmarks.
- Practical efficiency levers. The pipeline uses a small LVLM for perception (object/position Q&A) and a budget N to scale detail, giving users cost–quality control.

**Weaknesses:**

- Practical efficiency levers. The pipeline uses a small LVLM for perception (object/position Q&A) and a budget N to scale detail, giving users cost–quality control.
- Prompt dependence in question generation. The method hinges on a “powerful LLM” to craft good object/position prompts; robustness across domains or weaker LLMs isn’t deeply probed.

**Questions:**

Refer to the weaknesses,

---

> ### Author Response · Authors · 2025-11-21
> **Reponse to Reviewer wjkS**
>
> We sincerely thank the reviewer for the constructive comments, meticulous review and encouraging feedback. And we have conducted additional experiments and updated the manuscript accordingly. Our detailed responses are provided below.
>
> > **W1: Practical efficiency levers. The pipeline uses a small LVLM for perception (object/position Q&A) and a budget N to scale detail, giving users cost–quality control.**
>
> Thank you very much for acknowledging our contribution, and please feel free to discuss any further details regarding the pipeline.
>
> > **W2: Prompt dependence in question generation. The method hinges on a “powerful LLM” to craft good object/position prompts; robustness across domains or weaker LLMs isn’t deeply probed.**
>
> We thank the reviewer for raising this concern. In response, we explicitly evaluated the robustness of our question-generation component to both LLM size and domain shift. Using Qwen2VL-7B to produce initial captions for 500 natural images, we then generated questions with Qwen2-7B-Instruct and Qwen2-72B-Instruct under the same ScaleCap prompt. The accuracy of the generated questions is verified through a strict format matching process. Across domains, the question-format accuracy remained consistently high: for natural images, Qwen2-7B and Qwen2-72B achieved 97.5% and 98.7%; for Qwen2-72B on Infographic and STEM images (additional 500 images), accuracy reached 98.8% and 97.7%, respectively. This suggests that question generation is a relatively simple, robust task.
>
> | Instruction Generation Model | Domain | Instruction Accuracy |
> | :--- | :--- | :--- |
> | Qwen2-7B | natural | 97.5% |
> | Qwen2-72B | natural | 98.7% |
> | Qwen2-72B | Infographic | 98.8% |
> | Qwen2-72B | STEM | 97.7% |
>
> To further test end-to-end impact, we replaced the original instruction-generation model in ScaleCap with Qwen2-7B-Instruct while keeping all other components fixed and evaluated on the Prism subset. The average score drops slightly from 60.3 (Qwen2-72B) to 59.4 (Qwen2-7B), with MMVet/MMStar/ChartQA changing from 58.8/49.5/72.5 to 57.9/49.1/71.1. These results indicate that ScaleCap remains strong even with a weaker instruction-generation model, and that our method is not critically dependent on a single "powerful" LLM.
>
> | Instruction Generation Model | MMVet | MMStar | Chart QA | Avg |
> | :--- | :--- | :--- | :--- | :--- |
> | Qwen2-7B | 57.9 | 49.1 | 71.1 | 59.4 |
> | Qwen2-72B | 58.8 | 49.5 | 72.5 | 60.3 |
>
> Following your suggestions, we have included these results  in the Appendix and talk about LLM size effect on question generation.
>
> Thanks again for your constructive comments and meticulous review, which have helped us significantly improve the paper's quality! Please don't hesitate to let us know if there are any additional clarifications or experiments that we can offer.

---

> ### Author Response · Authors · 2025-11-27
> **Official Comment by Authors**
>
> Dear Reviewer wjkS,
>
> We sincerely appreciate your time and effort in providing such meticulous reviews and insightful comments.
>
> Could you please kindly let us know if our rebuttal has addressed your concerns?
>
> If you have any further questions regarding our work, we would be delighted to address them.
>
> Best regards,
>
> Authors

---

### Author Response · Authors · 2025-12-02
**Rebuttal Summary: Response to ALL Concerns**

**Dear Area Chair,**

Thank you for reviewing our submission. To facilitate a quick overview of our paper and the rebuttal status, we summarize below how we **have substantiated our responses to all concerns with extensive experimental evidence**.

## Reviewer wjkS (Initial Rating: 6)

* **Concern 1: Robustness of question generation.**
    * **Our Response:** Results demonstrate high stability and minimal dependence on **LLM size** (Instruction Accuracy: 7B 97.5% vs. 72B 98.7%) and **domains** (Instruction Accuracy: natural 98.7% vs. STEM 97.7%).

## Reviewer x2KT (Initial Rating: 6)
* **Concern 1: Studies about verbosity.**
    * **Our Response:** We conducted GPT-5 Quality Assessment. Results show ScaleCap matches baselines in Succinctness (ScaleCap 4.07 vs. Qwen2-VL-7B 4.05) while significantly outperforming them in Detail Coverage(ScaleCap 4.45 vs. Qwen2-VL-7B 3.51).

* **Concern 2: Robustness of the CSR threshold.**
    * **Our Response:** We conducted sensitivity analyses on the threshold parameter and cross-dataset evaluations, which demonstrate consistent performance (Precision ~65%, Recall ~73%) across varying thresholds and domains, confirming that CSR is robust and stable.

* **Concern 3: Used for SFT with ScalCap-450k.**
    * **Our Response:** We clarified that naive fine-tuning causes overfitting without access to the original training recipe, identifying this as a future direction.

* **Concern 4: The results with a different LVLM for HQA.**
    * **Our Response:** We successfully replaced the HQA model with InternVL3-8B. This setup improved the average score from 60.3% to 61.2%, demonstrating that ScaleCap can effectively leverage diverse LVLMs to capture broader visual information.

* **Concern 5: Algorithm Pseudocode.**
    * **Our Response:** We have added a structured pseudocode algorithm in the Appendix to enhance reproducibility and clarity.

## Reviewer jUSR (Initial Rating: 4)

* **Concern 1: Novelty of core components (CSR and Heuristic QA).**
    * **Our Response:** We clarified that our contributions lie in making unique observations about two major flaws in existing methods and designing new, specialized captioning paradigms to address them: (1) we designed the sentence-level Contrastive Sentence Rating (CSR) to resolve the 'fluency dilemma' observed in existing token-level methods; and (2) we developed Heuristic QA based on the insight that missing details stem from instruction constraints rather than perceptual blindness.

* **Concern 2: Computational cost and scalability.**
    * **Our Response:** We demonstrated ScaleCap is explicitly designed with a **tunable performance-budget trade-off** (Figure 7a) and emphasized the **offline nature** of dataset construction (a one-time investment for indefinite reuse).

* **Concern 3: Quantitative analysis of computational cost.**
    * **Our Response:** We provided the experiment results on 8 A100 GPUs: ScaleCap achieves a 2.2% Prism Score gain over Qwen2VL-72B with a 5x cost increase, whereas scaling from 7B to 72B yields only a 1.9% gain for a 15x cost increase. This demonstrates that ScaleCap effectively overcomes the diminishing marginal utility often seen in pure model scaling.

* **Concern 4: Performance comparison against training-based hallucination mitigation.**
    * **Our Response:** We compared ScaleCap against extensive training-based methods, including RLAIF-V, LLaVA-RLHF, HADPO, mDPO, and HALVA, which show that ScaleCap (training-free) outperforms most data-intensive training approaches (e.g., surpassing LLaVA-RLHF by ~13% on CHAIRS), ranking second only to RLAIF-V.

* **Concern 5: Potential benefits of integrating external tools.**
    * **Our Response:** We conducted new experiments by integrating Grounding DINO into the pipeline, which indicate marginal gains on natural images and performance drops on charts, primarily because (1) our CSR module already effectively filters hallucinations, and (2) generic tools lack robustness in specialized domains.

## Reviewer VAQo (Initial Rating: 4)

* **Concern 1: More evaluation metrics (e.g., GPT-based and QA-based metrics).**
    * **Our Response:** We incorporated ALOHa, CLAIR, Factuality, and Coverage, which cover all the suggested GPT-based and QA-based metrics. Results demonstrate that ScaleCap comprehensively outperforms baselines across all metrics (Avg Score: ScaleCap 74.1% vs. Qwen2VL-72B 72.2% vs. Qwen2VL-7B 70.0%).

* **Concern 2: Comparison with more baselines.**
    * **Our Response:** We conducted extensive experiments for comparison with CapMAS and VFC. Results show that ScaleCap comprehensively outperforms both methods (Avg Score: ScaleCap 74.1% vs. CapMAS 71.5% vs. VFC 70.8%).

---

### Meta-Review · Area_Chair_1Hyc · 2025-12-24

**Summary:**

The paper proposes a captioning pipeline centered around two components aimed at improving the generation of image captions using LVMLs (Large Vision Language Models): 1) heuristic question answering and 2) contrastive sentence rating. The former generates content-specific questions based on the image and answers them to progressively inject relevant information into the caption. The latter employs sentence-level offline contrastive decoding to effectively identify and eliminate hallucinations caused by linguistic biases. The method allows for scaling budget controls to trade off computation costs and caption content detail evaluated. The proposed pipeline is used to generate a dataset which enable continuous pretraining demonstrating consistent performance gains on 11 widely used benchmarks for evaluating visual language models.

The paper has been indicated as well-written and structured, and has been praised for proposing a clear modular method that specifically targets two real failure modes in captioning models. Reviewers also appreciated the extensive empirical evaluation which includes demonstrating the effectiveness of a pretraining pipelines on 11 benchmarks.

The main weaknesses identified by reviewers relate to the fact that the method primarily combines and refines already existing techniques without introducing fundamentally new architectural components. In addition, reviewers were on the fence about the resource requirements involved in the deployment of the method that could limit the scalability.

Overall though, the controllability of the pipeline afforded by its modularity and the possibility given to the user to control the trade-off between computation cost and caption detail seem to at least partially counterbalance both of these main concerns, resulting in an architecture of high potential in terms of practical applicability.

**Reviewer Concerns:**

* Addressed in the rebuttal:
  - Effect of caption length and sheer verbosity in observed gains have been addressed by quantifying verbosity using GPT-5 and showing that the proposed approach achieves verbosity (succinctness scores) comparable to the Qwen2-VL 7B and 72B, while achieving higher detail and accuracy
  - Comparisons with existing methods for improving caption detail have been added in the rebuttal with empirical proxy comparisons (absent official public implementations). These experiments show that ScaleCap outperforms methods like CapMAS and VFC.
  - Sensitivity analyses on the threshold parameter and cross-dataset evaluations demonstrate robustness of the proposed method across varying parameters
  - Requested pseudocode implementing the method has been provided in the revised version
  - Additional GPT-based and QA-based evaluation metrics requested by reviewers have been added in the rebuttal, confirming the advantages of the proposed method

* Not addressed in the rebuttal:
  - The method hinges on a “powerful LLM” to craft good object/position prompts. The rebuttals proposed some ablation studies with smaller LLMs, but results seem to only partially address the concern (e.g. Qwen2-7B performance was only shown for natural domain)
  - Limited novelty concerns raised by reviewers have been only partially addressed in the rebuttal
  - Concerns regarding high computational cost of the proposed method not fully addressed, although as noted in the rebuttal the method allows for trading off computation and caption detail which is also parallelizable, and the main scope of the work is to improve accuracy

**Reviewer Scores:**

| Reviewer | initial score | predicted final score |
|---:|---:|---:|
| wjkS | 6 | 6 |
| x2KT | 6 | 6 |
| jUSR | 4 | 4 |
| VAQo | 4 | 4 |

---

### Decision · Program_Chairs · 2026-01-26

Accept (Poster)